# Piper: Multidimensional Planner
# for DNN Parallelization

**Jakub Tarnawski**
Microsoft Research
jakub.tarnawski@microsoft.com

**Deepak Narayanan**
Microsoft Research
dnarayanan@microsoft.com

**Amar Phanishayee**
Microsoft Research
amar@microsoft.com

## Abstract

The rapid increase in sizes of state-of-the-art DNN models, and consequently the increase in the compute and memory requirements of model training, has led to the development of many execution schemes such as data parallelism, pipeline model parallelism, tensor (intra-layer) model parallelism, and various memory-saving optimizations. However, no prior work has tackled the highly complex problem of optimally partitioning the DNN computation graph across many accelerators while combining *all* these parallelism modes and optimizations. In this work, we introduce Piper, an efficient optimization algorithm for this problem that is based on a two-level dynamic programming approach. Our two-level approach is driven by the insight that being given tensor-parallelization techniques for individual layers (e.g., Megatron-LM's splits for transformer layers) significantly reduces the search space and makes the global problem tractable, compared to considering tensor-parallel configurations for the entire DNN operator graph.

## 1 Introduction

Deep Neural Network (DNN) models have grown exponentially in size over the past two decades; consequently, modern DNNs are extremely computationally expensive. For example, a state-of-the-art language model, GPT-3 [2], has an astounding 175 billion parameters, requiring 314 ZettaFLOPS ($3.14 \times 10^{23}$ FLOPS) of compute.

This trend of computationally expensive models with working-set sizes for training and inference that far exceed the memory capacities of individual DNN accelerators (e.g., GPUs, TPUs, FPGAs, ASICs) has reinvigorated the study of parallel training techniques that split DNN model state across different devices. In the "modern era", such model-parallel training techniques trace their roots back to AlexNet [14] and early influential systems such as DistBelief [6] and Project Adam [3]. Recent advances in model-parallel training include *tensor (intra-layer) model parallelism* [22], where individual operators of the model are partitioned over multiple workers, and pipeline model parallelism that combines inter-layer model parallelism with pipelining [17, 18, 11]. Various hybrid parallelism techniques that combine tensor, pipeline, and data parallelism across operators [13, 19] have also been developed to more efficiently train a wider range of models. Additionally, memory-saving optimizations like activation recomputation, which trade off throughput for memory, are also being used [18] to scale training to larger models such as GPT-3.

Combining these dimensions, however, is non-trivial [19], since each dimension has trade-offs with respect to computational efficiency, amount of communication, and memory footprint. Given the importance of *efficient* model-parallel training (and inference [7, 4]), partitioning a model across

35th Conference on Neural Information Processing Systems (NeurIPS 2021).

devices has thus received recent interest, evolving from a manual process driven by human experts to using algorithms for automated device placement of DNN operators. Focusing on algorithms, some take a black-box approach using learning-based techniques [16, 15, 8, 1, 26, 21, 25]; others build a cost model and algorithmically solve an optimization problem [12, 17, 18, 13, 23]. However, each of the prior algorithmic approaches tackle a slightly different subset of parallelization techniques, system constraints, and DNN optimizations; for example, FlexFlow [13] does not consider pipelining, PipeDream [17] does not consider memory-saving optimizations that are useful for training large models, and PipeDream and PipeDream-2BW [18] both do not consider tensor parallelism.

**Our contribution.** In a first, this paper covers a broad search space of parallelization and optimization techniques while determining placement of DNN layers on the provided hardware deployment (a given number of accelerators of a specific type with certain performance characteristics, memory capacity, and interconnect bandwidth). Specifically, our optimization algorithm, called Piper, considers three different dimensions of parallelism (data, tensor model, and pipeline model parallelism), and a general framework of memory-saving optimizations that trade off throughput for lower memory footprint. In this work, we consider pipeline parallelism schedules without pipeline flushes [17, 18]. We leave extensions to pipeline parallelism schedules with flushes [11, 19] to future work.

The focus of our work is on algorithms for DNN partitioning across devices to maximize throughput (for training and large-batch inference of DNN models). Piper uses non-trivial combinations of data, tensor, and pipeline parallelism, as well as memory-saving optimizations such as activation recomputation, carefully chosen for each stage of the partition. Piper supports general DAG (Directed Acyclic Graph) model topologies and is agnostic of the type of DNN accelerator used. Piper is given a target "safe" batch size (e.g., 1920 for large GPT-style models [20]) and returns a training configuration with this batch size. It is efficient both in theory and in practice.

In Section 5, we evaluate Piper on real-world DNN profiles, and study the effects of combining the various parallelism modes and memory-saving optimizations on performance. We find that Piper's large search space combined with its principled algorithmic approach allows it to find non-trivial high-quality parallelization configurations for each given problem instance. Piper compares favorably to planners from prior work (PipeDream, PipeDream-2BW). For example, we find (as expected) that in many settings, pipeline model parallelism combined with per-shard data parallelism is enough to obtain high throughput; however, tensor parallelism is also required when one needs a batch size smaller than the number of devices, as well as in highly memory-constrained settings. Piper's ability to form heterogeneous stages also gives it an advantage over prior work.

## 2  Background

In this section, we briefly describe methods to train models at scale.

### 2.1  Pipeline model parallelism

Pipeline model parallelism is a recent technique [17, 11] used to accelerate the distributed training of both small and large models. The operators of a model are partitioned over the available devices (i.e., each device hosts a subset of the operators of the model; typically these subsets are contiguous). A batch is then split into smaller microbatches, and execution is pipelined across these microbatches: while the second device starts executing the first microbatch, the first device starts executing the second microbatch, and so on. Pipeline model parallelism features cheap point-to-point communication.

A key problem with deploying pipeline parallelism is determining how to schedule forward and backward microbatches on the available devices, and how to stash state to ensure a final high-quality model. Different schedules have different throughput, memory footprint, and semantics trade-offs. For example, GPipe [11] maintains just a single weight version, but introduces periodic pipeline flushes where microbatches in the pipeline are allowed to drain. This leads to idle resources and lower throughput, but also lower memory footprint. PipeDream [17] maintains a weight version for every in-flight input microbatch, but does not use pipeline flushes, leading to both high throughput and memory footprint. PipeDream-2BW [18] tries to find a middle ground, where gradients are coalesced across microbatches, reducing the number of weight versions, while still keeping throughput high. In this work, we assume that a non-flushing pipelining scheme is used when considering pipelines with multiple stages (i.e., we do not use inter-layer model parallelism without pipelining).

## 2.2 Tensor (intra-layer) model parallelism

A model can be partitioned in other ways as well. Megatron-LM [22] slices transformer blocks in NLP models like BERT and GPT across available devices, in effect performing distributed matrix multiplications. All-to-all communication in the form of scatters, gathers and all-reductions needs to be performed to ensure that these distributed operators retain their semantics. Megatron-LM uses a specific, hand-tuned strategy to distribute transformer layers across multiple GPUs. Tensor parallelism reduces per-device memory usage, as model parameters are split across devices, and shows good scaling performance when using high-bandwidth interconnects like NVLink [19].

Trying to find an optimal tensor parallelization strategy for the entire model by deciding how to split every operator is a highly challenging problem with a large search space, and requires understanding the semantics of each operator. Modern models consist of a large number of operators, often with considerable branching. Moreover, each operator can be split in many different ways. For example, a matrix multiplication can be split over the rows or columns of the left (input) matrix or right (weight) matrix, or a combination of these. Some splits will require inserting an all-reduction operator to recover the result, and some will require an all-gather. Some will divide weights across devices, which helps with memory usage. The optimal split in terms of latency depends on the matrix dimensions and also downstream operators; a matrix multiplication split over some dimension would accept input tensors split over the same dimension, but a dropout operator requires the entire input tensor.

## 2.3 Data parallelism

Data parallelism is the de-facto way of parallel training today. With data parallelism, every worker has a copy of the full model. Inputs are sharded, and each worker computes weight gradients independently. Periodically, these weight gradients are aggregated using all-to-all communication or a parameter server. Vanilla data parallelism assumes that model parameters fit on a single device. For large models, data parallelism can be used on model shards; gradients need to only be exchanged among devices responsible for the relevant model shard. Data parallelism can be used with both pipeline (as shown in PipeDream [17]) and tensor model parallelism (as shown in Megatron-LM [22]).

## 2.4 Memory-saving optimizations

When training large models, it is common to use memory-saving optimizations that trade off a reduction in throughput for a lower memory footprint. One example is **activation recomputation**, where the intermediate activations produced during the forward pass are not stored. Instead, the forward pass is performed a second time, just before the backward pass, from a stashed *input* activation, which is much smaller than the full set of intermediate activations. This drastically reduces the footprint of pipelined training, at the cost of executing the forward pass twice.

## 2.5 Related work on algorithms for partitioning and placement of DNN graphs

A number of systems have looked at the problem of partitioning DNN models across devices. However, each of these systems considers a subset of the parallelism dimensions considered in this paper. FlexFlow [13] considers many dimensions (data, tensor, and inter-layer model parallelism), but not pipelining; moreover, it uses a heuristic MCMC-based approach that requires domain knowledge to be encoded into the "parallel plan" generator. PipeDream [17] and Tarnawski et al. [23] use dynamic programming and consider a combination of pipeline model parallelism and data parallelism; however, they support different model topology classes. PipeDream supports only linear (path) graphs while [23] supports general DAGs. While PipeDream is oblivious to memory usage, its enhancement, PipeDream-2BW [18], targets large models that do not necessarily fit on a single accelerator. Exploiting the repetitive structure of some of these large models, such as transformer-based language models, PipeDream-2BW's planner only considers configurations where every stage in the pipeline is replicated an equal number of times (equi-replicated stages), thus simplifying the search for parallelization configurations. PipeDream-2BW's planner also explicitly considers per-device memory capacity constraints and includes activation recomputation in its configuration search space. None of these three approaches consider tensor parallelism.

Learning-based approaches (Mirhoseini et al. [16, 15], Spotlight [8], Placeto [1], GDP [26], RE-GAL [21]) treat the objective (latency or throughput) as a black box to be optimized using RL;

runtimes are measured in an online fashion on the target system, which is very accurate but computationally expensive. Moreover, some of them optimize for latency or peak memory usage rather than throughput, and they do not explicitly consider splitting individual operators or pipelining.

All existing algorithms, except PipeDream-2BW, also do not consider memory-saving optimizations, that trade off throughput for reduction in memory footprint, in combination with different modes of parallelism; activation recomputation is one such important optimization. These optimizations are commonly needed for training models that exceed the memory capacities of modern accelerators.

## 3 Problem Setup

In this section, we describe the search space of parallelization configurations that Piper explores, as well as the objective function (cost model) that we minimize in order to find the best solution. These give rise to the description of the optimization problem that Piper solves.

Piper is given as input a DNN workload (model) which is represented as a Directed Acyclic Graph (DAG), $G = (V, E)$. Each node corresponds to a single DNN layer such as LSTM or transformer (each layer is composed of lower-level operators, e.g., matrix multiplications or additions). Each edge $(u, v)$ corresponds to a data transfer: node/layer $v$ requires the output of $u$ as its input. The input graph is annotated with a number of attributes related to compute times, communication costs, and memory usage; we introduce and motivate these attributes in this section, and then formally list them in Section 3.6. To obtain an input to be supplied to Piper, these attributes should be profiled (or estimated). Furthermore, Piper is given: the number of devices ($K$), available memory per device ($M$), the network bandwidth ($B$), and the target number of microbatches in a batch ($N$). $N$ is the ratio of the chosen batch size (usually the maximum that is safe for convergence, e.g., 1024–2048 for large transformer-based LMs) to the provided microbatch size.

### 3.1 Space of configurations

Piper's search space consists of partitions of $G$ into consecutive contiguous subgraphs, also called stages, which are executed using disjoint sets of devices. Each node/layer is assigned to exactly one stage. Each stage is executed using some number $d \cdot t$ of devices, combining data parallelism of degree $d$ with tensor parallelism of degree $t$. That is, we want to find a partition $V = S_1 \cup ... \cup S_\ell$ and, for each stage $i = 1, ..., \ell$: **(a)** the degree $d_i$ of data parallelism, **(b)** the degree $t_i$ of tensor parallelism, **(c)** further configuration: how (for each layer in the subgraph) the tensor parallelism is carried out, as well as what other optimizations are used.

Thus the $i$-th stage will be run on $d_i \cdot t_i$ devices, and so we must have $\sum_{i=1}^{\ell} d_i \cdot t_i \leq K$. Furthermore, each stage should be feasible in terms of memory usage, the sum of data-parallel degrees ($\sum_{i=1}^{\ell} d_i$) must be at most $N$ (in order to saturate the pipeline), and we need $d_i$ to divide $N$ (to be able to divide microbatches evenly among the data-parallel replicas). We call this output a **solution**. We use the definition of contiguity from [23] (see Fig. 6 in the Appendix for an illustration):

**Definition 3.1** *We say that a set $S \subseteq V$ is* contiguous *if there do **not** exist nodes $u \in S$, $v \in V \setminus S$, and $w \in S$ such that $v$ is reachable from $u$ and $w$ is reachable from $v$.*

### 3.2 Time-Per-Sample

Our objective is to maximize throughput; equivalently, we minimize its inverse, the **Time-Per-Sample (TPS)**. For non-flushing pipeline execution schemes such as PipeDream or PipeDream-2BW, it has been argued [23] as well as shown in practice [17] that the Time-Per-Sample of a solution equals the maximum Time-Per-Sample of a device. The problem of finding a solution with minimal TPS (absent data or tensor parallelism and other optimizations) was addressed by Tarnawski et al. [23]; in that model, which we extend, the TPS of a device/subgraph is given as the sum of compute times of all nodes in the subgraph, plus costs of communication along edges coming into or leaving the subgraph. We now discuss how to incorporate the modes of parallelism from Section 2 into the model.

### 3.3 Tensor Model Parallelism Configurations (TMPCs)

As we remarked in Section 2.5, the task of making a global, joint decision about how to tensor-parallelize every *operator* in a DNN computation graph is very complex. In this work we therefore **reduce** this task to the problem of coming up with good tensor parallelization techniques *for individual layers*. Solving the latter problem is not a goal of Piper; instead, we assume that we already know (i.e., Piper is given as input) some number of good tensor parallelism configurations (for some layers $v$ and some degrees $t$ of tensor parallelism). In practice, such configurations can be devised by hand (as done e.g. for transformer layers in Megatron-LM [22]; we use their technique in our experimental evaluation), or the problem can be attempted using automated methods similar to e.g. FlexFlow [13] or Tofu [24] or RL-based approaches. This problem has to only be solved for each layer *type*, e.g., even if the input DNN has many transformer layers, one just needs a good technique for *a transformer layer*. It is also not necessary to provide configurations for all layers $v$ and possible degrees $t$. On the other hand, there could be many reasonable tensor parallelism configurations for the same $v$ and $t$; for example, there might be a trade-off between memory usage and runtime.

For each available configuration, Piper gets as input certain attributes related to compute, communication and memory costs for the target model (listed formally in Section 3.6). A tuple containing these attributes is called a *Tensor Model Parallelism Configuration (TMPC)*. Piper receives a (possibly empty) list $T(v, t)$ of TMPCs for each node $v$ and degree $t$ of tensor parallelism (including $t = 1$). Memory-saving optimizations (e.g., activation recomputation) are modeled using TMPCs as well.[1]

### 3.4 Data parallelism

Consider a stage that is data-parallelized across $d > 1$ devices. Following PipeDream [17], we model the effect of this on TPS as follows. The compute load of the stage is spread evenly across devices (with a factor $1/d$). However, data-parallel execution necessitates periodic synchronization of weight gradients, which has a total communication cost of $4(d-1)w$, where $w$ is the size of parameters in all layers in the stage. This gives a cost of $4 \cdot \frac{d-1}{d} \cdot w$ bytes per device (per batch, i.e., $N$ microbatches).

### 3.5 Memory usage in pipelined configurations

With pipelining, every stage needs to store some number of weight and activation versions; these depend on the number of in-flight microbatches (per data-parallel replica). Specifically, the memory usage of a stage can be modeled as an **affine function** of this number. That is, for every stage with data-parallel degree $d$ and suffix sum of data-parallel degrees $s$, memory usage would be $a \cdot \lceil s/d \rceil + b$ for appropriate coefficients $a$ and $b$. See Appendix A for an explanation. E.g., in PipeDream-style execution, $a$ would be the size of weights plus the size of all activations in the subgraph, and $b$ would be the size of optimizer state and temporary buffers. PipeDream-2BW uses only two stashed weight versions instead of $\lceil s/d \rceil$ many, so the size of weights would contribute only to $b$ (multiplied by 2), not $a$. If activation recomputation is used, then $a$ accounts for only the sizes of *input* activations of every layer, and $b$ accounts for all other activations. This accounting is more precise than PipeDream's, which ignores memory usage, or Tarnawski et al.'s [23], which uses a constant function.

### 3.6 Input and output

Now we can formally (re-)state Piper's **input**:

- a Directed Acyclic Graph $G = (V, E)$: layer graph of a DNN,
- for every edge $(u, v) \in E$, an associated communication cost $c(u, v)$ (in bytes),
- number $K$ of devices, memory $M$ per device, network bandwidth $B$, [maximum] number $N$ of microbatches in a batch,
- for every node/layer $v \in V$, and every degree $t = 1, ..., K$ of tensor parallelism: a (possibly empty) list $T(v, t)$ of TMPCs (see Section 3.3).

The description of a TMPC in $T(v, t)$ must correspond to a tensor parallelism configuration where, for both the forward and the backward pass, each of the $t$ devices start execution already with a full

---

[1]As an example, a node $v$ might have the following TMPCs as input: for $t = 1$, one TMPC corresponding to regular execution, and another with activation recomputation; and for $t = 8$, one TMPC corresponding to tensor parallelization in dimension 1 and another using dimension 2, with and without activation recomputation (four in total).

copy of each input activation, and must end execution having in its memory a full copy of every output activation (see Appendix B for a discussion). Each TMPC description $X \in T(v, t)$ contains:

- a Time-Per-Sample $X.p$ that encompasses the compute time together with the communication costs of tensor parallelism (data transfers between the $t$ devices),[2]
- for every incoming edge $(u, v) \in E$, a communication cost $X.c^{\mathsf{fw}}(u, v)$ (in bytes) that it would take to synchronize the tensor incoming over that edge between $t$ devices (going from a state where one device has the tensor to a state where all $t$ have it); for more explanation and motivation of this, see Appendix B,
- an analogous quantity $X.c^{\mathsf{bw}}(v, w)$ for every outgoing edge $(v, w) \in E$,
- the size $X.w$ (in bytes) of parameters/weights on each device,
- memory usage: two amounts $X.a$, $X.b$ of bytes such that if layer $v$ is located on a stage with data-parallel degree $d$ and data-parallel degrees of this and later stages sum up to $s$, then the memory consumption on each device is $X.a \cdot \lceil s/d \rceil + X.b$ (see Section 3.5).

Piper **outputs** the following low-TPS solution: a collection of contiguous subgraphs/stages $S_1, ..., S_\ell$ such that $V = S_1 \cup ... \cup S_\ell$, and for each stage $i$: **(a)** the degree $d_i$ of data parallelism, **(b)** the degree $t_i$ of tensor parallelism, **(c)** for each node/layer $v \in S_i$: the index of the TMPC selected in $T(v, t_i)$.

All nodes in a stage use the same degrees of data and tensor parallelism, but different nodes (even if all correspond to e.g. transformer layers) can use different TMPCs. For example, some but not all layers in a stage might employ activation recomputation.

## 4   Algorithm

Our algorithm is based on dynamic programming on *downsets*, extending recent work [23].

**Definition 4.1** *We call a set $D \subseteq V$ of nodes a* downset *if $(u, v) \in E$ and $u \in D$ implies $v \in D$.*

As proved there, going from downset to downset we can obtain every possible contiguous subgraph:

**Fact 4.2** *A set $S \subseteq V$ of nodes is contiguous (see Definition 3.1) if and only if it is the difference of two downsets: $S = D \setminus D'$ where $D' \subseteq D$.*

We rely on the domain insight that layer-granularity computation graphs of modern DNNs do not have many downsets, making tractable the approach of considering all downsets and all contiguous sets using dynamic programming. We define the following **dynamic programming table**: $dp[D][k][s]$ is the minimum TPS of a solution that partitions the downset $D$ among $k$ devices (recall that the TPS of a solution is the maximum TPS over all $k$ devices) such that the sum of data-parallel degrees of all stages is $s$. A key novel point in our approach is that using the single variable $s$, which ranges from $0$ to $N$, we can precisely control both memory usage (see Section 3.5) and batch size (Appendix A).

We **initialize** the table as follows: $dp[\emptyset][k][s] = 0$ for every $k$ and $s$. The answer (minimum TPS) when using any number $k$ of devices and sum $s$ of data-parallel degrees will be held in $dp[V][k][s]$, and the final answer is $\min_{k=1,...,K} \min_{s=1,...,N} dp[V][k][s]$.

We proceed to the main part: **the recursion**. For every possible sub-downset $D'$, we consider partitioning the contiguous subgraph $S = D \setminus D'$ with data-parallel degree $d$ and tensor-parallel degree $t$, so that $d \cdot t$ devices are used for $S$ and the rest for $D'$. For $\emptyset \subsetneq D \subseteq V$, we have

$$dp[D][k][s] = \min_{\text{downset } D' \subseteq D} \; \min_{\substack{d=1,...,s \\ d \mid N}} \; \min_{t=1}^{\lfloor k/d \rfloor} \max(dp[D'][k - d \cdot t][s - d], \text{TPS}(D \setminus D', t, d, s)),$$

where we **define** $\text{TPS}(S, t, d, s)$ as the minimum Time-Per-Sample when optimally partitioning $S$ with degrees $d$ and $t$ of data and tensor parallelism, respectively, and so that the sum of data-parallel degrees of this and all further stages is $s$. We now describe how we compute this quantity.

**The knapsack subproblem.**   How to compute $\text{TPS}(S, t, d, s)$? Given parallelism degrees $d$ and $t$, we need to choose one TMPC $X \in T(v, t)$ for each $v \in S$. Some TMPCs may have better

---

[2]Includes the runtime of both the forward and the backward passes.

TPS, but others may use less memory. More concretely, we want a combination that minimizes the total TPS of the stage while staying under the memory limit $M$. Both the total TPS and the total memory usage of any combination can be expressed as sums of the TPSes and the memory usages of TMPCs of individual nodes $v \in S$ (more details below). However, there are still exponentially many $\left(\prod_{v \in S} |T(v, t)|\right)$ possibilities, and in fact this subproblem is NP-hard, as it generalizes the 0-1 knapsack problem. At the same time, we require a very fast solution as it needs to be solved for all $S$, $t$, $d$, $s$. We use a simple bang-per-buck heuristic:

- start by picking the lowest-TPS (fastest) TMPC for each $v \in S$,
- as long as the memory usage is too high, pick the TMPC in $\bigcup_{v \in S} T(v, t)$ whose ratio of decrease in memory usage (compared to the currently picked TMPC for $v$) to increase in TPS is the highest, and pick it for $v$ (instead of the currently picked one).

If no combination of TMPCs is memory-feasible, we return $+\infty$. This is the only non-exact component of the algorithm; however, we find that this heuristic almost always finds the optimal solution in practice. See Appendix C for more discussion.

**TPS of a fixed configuration.** Finally, how do we compute the TPS of a stage for given $S$, $t$, $d$, $s$ and a selection of an TMPC $X(v) \in T(v, t)$ for every $v \in S$? We need to take compute, communication and memory into account. **Compute** is the simplest: compute $= \frac{1}{d} \sum_{v \in S} X(v).p$. For **communication**, we have costs stemming from sending activations into $S$, sending activations out of $S$, and costs of all-reductions from using data parallelism (all in bytes):

$$
\text{comm} = \frac{1}{d} \left( \sum_{(u,v) \in \delta^-(S)} 2 \left( c(u,v) + X(v).c^{\text{fw}}(u,v) \right) \right.
$$
$$
\left. + \sum_{(v,w) \in \delta^+(S)} 2 \left( c(v,w) + X(v).c^{\text{bw}}(v,w) \right) \right) + 4 \cdot \frac{d-1}{d \cdot N} \cdot \sum_{v \in S} X(v).w
$$

where we used the notation $\delta^-(S)$ and $\delta^+(S)$ for the sets of incoming and outgoing edges of $S$. **Memory usage** is given by: $\text{mem} = \sum_{v \in S} \left( X(v).a \cdot \lceil s/d \rceil + X(v).b \right)$. The final TPS for the selected $S$, $t$, $d$, $s$ and $\{X(v)\}_{v \in S}$ is compute $+ \frac{\text{comm}}{B}$ if mem $\leq M$; otherwise we return $+\infty$.

**Running time.** Under some natural assumptions, Piper's running time is $\tilde{O}(|V|^2 N(|V| + K \cdot d(N)))$, where $d(N)$ is the number of divisors of $N$. A detailed analysis can be found in Appendix D.

## 5   Evaluation

We now evaluate the Piper algorithm on real-world DNN models. Besides the efficiency and scalability of Piper, we are also interested in the advantages of the configurations produced by Piper over best possible configurations that forgo at least one of the modes of parallelism or memory-saving optimizations, as well as over best possible *equi-partitioned configurations* such as those found by PipeDream-2BW's planner algorithm. We stress that we focus on evaluating the Piper *partitioning algorithm* in this work, not any particular pipelined DNN training system. Moreover, with a fixed batch size and pipelining scheme, minimizing TPS is equivalent to minimizing time-to-convergence.

**Inputs.** For our comparisons, we use a BERT-32 model, which consists of an embedding layer, 32 transformer layers and a pooling layer. This model has 5.9 billion parameters, with similar dimensions to the GPT-3 model with 6.7B parameters [2], and does not fit on a single GPU. We provide TMPCs for non-tensor-parallelized ($t = 1$) and tensor-parallelized executions of transformer layers [22] ($t \in \{2, 4, 8\}$), each with and without activation recomputation. These TMPCs are obtained by profiling models implemented in PyTorch on NVidia A100 GPUs interconnected with a 300 GB/s bandwidth NVSwitch within a server, and 25 GB/s across servers. In our evaluation, we assume that all $K \leq 2048$ devices are connected by a flat network topology with a (full bisection) bandwidth of $B = 25$ GB/s (used for all non-tensor-parallelism communication). Our TMPCs model the memory footprint assuming the PipeDream-2BW pipelining scheme.

**Baselines.** We compare Piper to:

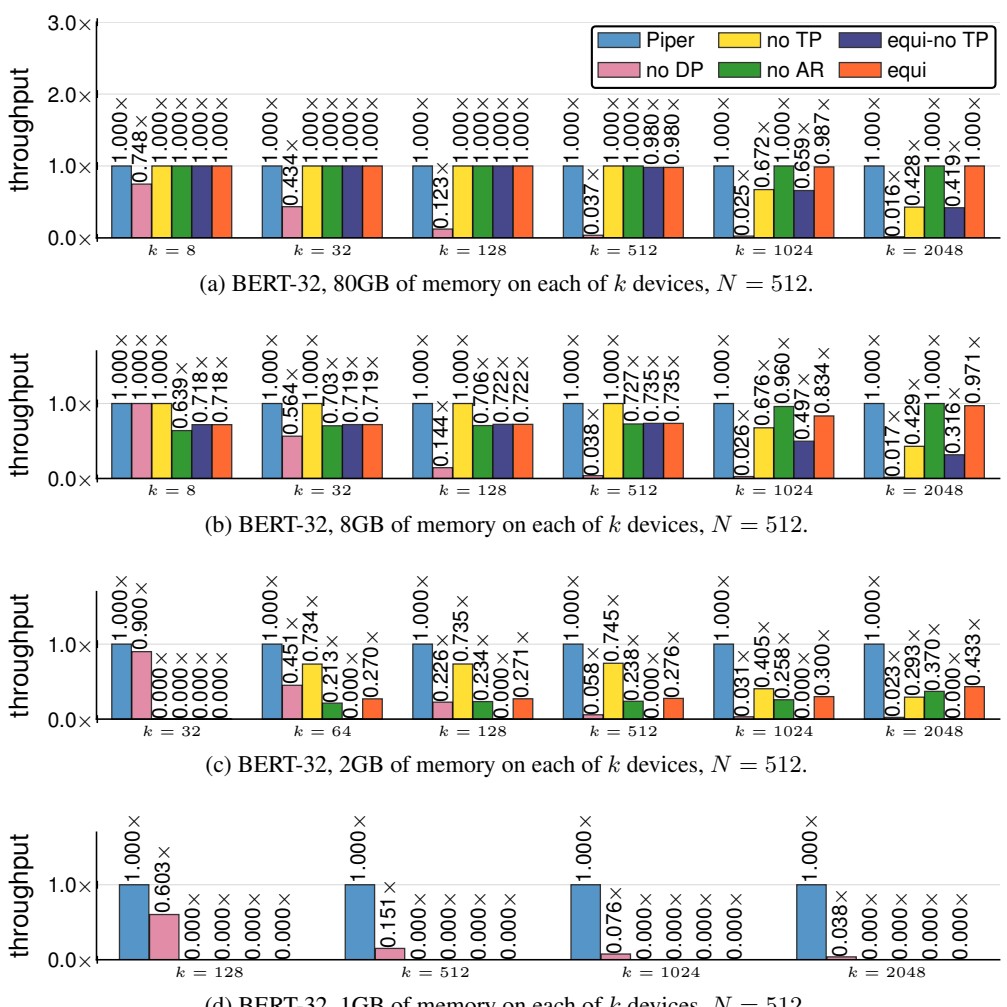

Figure 1: Comparison of Piper to baselines on the basis of throughput normalized to the best configuration. Throughput is the inverse of TPS (Time-Per-Sample); higher throughput is better. Piper always obtains the highest throughput ($1.000\times$). We vary the numbers of devices (from $8$ to $2048$) and the memory per device, which impacts the space of feasible solutions. For 1GB- or 2GB-devices, baselines are often unable to find *any* memory-feasible configuration ($0.000\times$). 1GB models a scenario where TP and AR are both required to fit even a single layer on a device.

- **no DP**: a version of Piper that forgoes data parallelism (i.e., $d = 1$),

- **no TP**: a version of Piper that forgoes tensor parallelism (i.e., $t = 1$),

- **no AR**: a version of Piper that forgoes activation recomputation,

- **equi**: an algorithm that equi-partitions layers into $w$ equal groups and uses $d$-data parallelism and $t$-tensor parallelism, choosing the lowest-TPS among all valid $(w, d, t)$ triples; it also considers performance with and without activation recomputation (for the entire model), and places the embedding and pooling layers either on separate stages or with the first and last transformer layer respectively,

- **equi-no TP**: a version of **equi** that forgoes tensor parallelism (i.e., $t = 1$); we show this as a surrogate for PipeDream-2BW's planning algorithm.

Both **no TP** and **no AR** are more sophisticated than PipeDream's partitioning algorithm, which forgoes tensor parallelism, activation recomputation, and also assumes $M = \infty$.

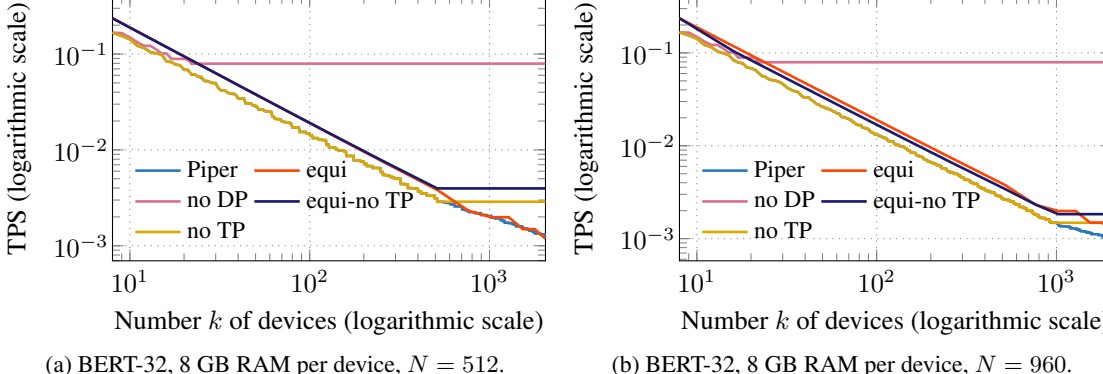

(a) BERT-32, 8 GB RAM per device, $N = 512$.    (b) BERT-32, 8 GB RAM per device, $N = 960$.

Figure 2: A *single run* of Piper produces a table of optimal Times-Per-Sample (TPS) for every number of devices. Here, we show the best TPS for Piper and the baselines versus the number of devices, for two different batch size ($N$) values. Lower TPS is better. Tensor parallelism is needed to use more than $N$ devices, as seen by the **no TP** baseline plateauing after $k = N$. **equi** obtains a 10-15% worse TPS than Piper for most $k$.

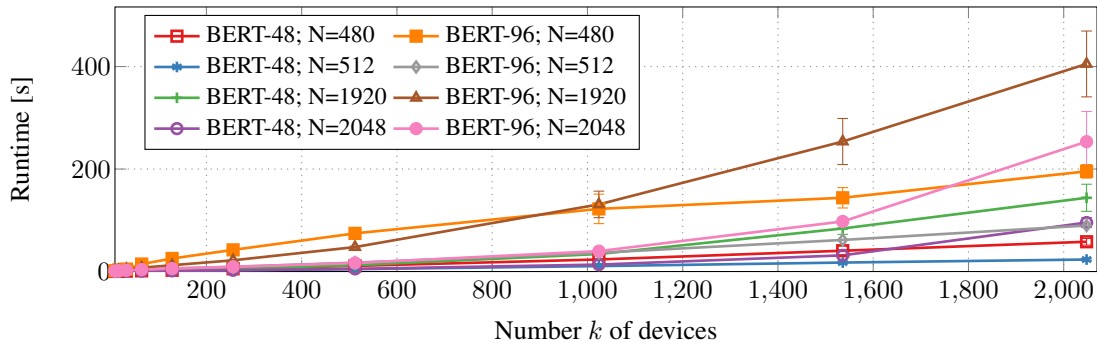

Figure 3: Piper's running time as a function of the number of devices, for various DNN model sizes and batch sizes ($N$). Bars show standard deviation. Single-core implementation executed on an Intel Xeon E5-2673 v4 CPU. Runtime depends on the number of divisors of $N$ (over which $d$ is iterated).

## 5.1 Results

Results of our evaluation in terms of the quality (TPS) of the obtained configurations are given in Figs. 1 and 2. The running times of Piper are shown in Fig. 3. We discuss the main takeaways.

**Data parallelism (DP).** Data parallelism is crucial, as expected. Without it, scaling to more devices is impossible; e.g., in Fig. 2, the TPS of **no DP** stops decreasing beyond $k = 22$ devices.

**Tensor model parallelism (TP).** We found that, in a vacuum, the speedup obtained by TP is much less than that of DP due to frequent and costly synchronization. However, TP also shrinks the parameter size per device, which helps with memory feasibility, and does not increase $s$ (in contrast to DP), which enables one to stay below the specified number $N$ of microbatches in a batch. In particular, TP is essential to use $k > N$ devices (see the **no TP** baseline in e.g. Fig. 1a or Fig. 2). TP is also required in cases when even a single layer cannot fit onto a device (Fig. 1d).

**Activation recomputation (AR).** Activation recomputation is required to obtain *any* memory-feasible solution in certain cases. Even when it is possible to do so without AR, foregoing it can force quite imbalanced (in terms of TPS) splits in order to fit in memory. For example, in Fig. 1c, without AR, one is forced to use small values of $s$ to stay within memory, which limits the number of stages and data-parallel degrees, and hence performance (using TP as a replacement yields worse throughput). In scenarios with enough memory relative to DNN size (Fig. 1a), AR is not useful.

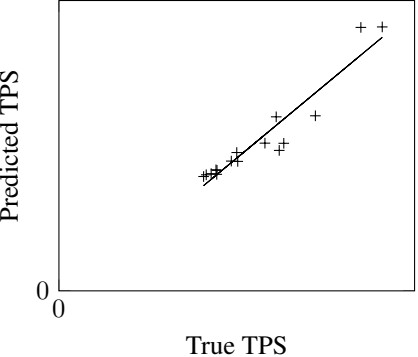

Figure 4: Real vs. Piper-predicted Time-Per-Sample for 16 configurations of pipeline, data, and tensor parallelism for BERT-32 with $K = 64$ devices. Training times were measured on a system with 8 NVidia DGX A100 machines, each with 8 GPUs. The Pearson correlation coefficient is 0.95.

**Power of Piper's search space.** Piper has the ability to return configurations with stages that are different from each other (e.g., different numbers of layers, degrees $d$ and $t$, memory optimizations). We found that this leads to a lower optimal TPS than the baseline **equi** even for extremely repetitive DNNs such as BERT (see Figs. 1 and 2). This is useful for two reasons. First, in earlier stages, the parameter $s$ that controls memory usage (Section 3.5) is higher (as $s$ is a "suffix sum" of DP degrees over stages), which means that it is optimal to have larger degrees $d$ or $t$ or smaller numbers of layers in the earlier stages. In other words, in an optimal *equi-partitioned* configuration, if memory is a bottleneck, then the memory limit will be attained almost tightly in the first stage, which needs to store activations for many in-flight microbatches, but will be loose in the last stage, which only stores activations for one microbatch. In contrast, Piper will effectively adjust all stages to ensure that almost all available memory is used on every device. Second, Piper is also able to adjust to different layer types; for instance, the embedding layer can use a different TP degree $t$ than transformer layers.

**Running time of Piper.** Fig. 3 shows the scalability of Piper's algorithm as a function of $k$ and $|V|$, as well as $N$, on larger BERT inputs created by expanding our BERT-32 profiles. We find that Piper terminates in minutes, which is a very small percentage of the actual training times of the DNNs (days to weeks). We also note that Piper's algorithm is embarrassingly parallel, as for every $I$, computing the table $\text{TPS}(I \setminus I', \cdot, \cdot)$ and updating $dp$ can be done in parallel over all sub-downsets $I' \subseteq I$. With such an implementation, runtimes for the algorithm would scale linearly on a multi-core CPU server.

**Quality of cost model.** Fig. 4 shows that Piper's cost model is very close to real performance. In this work, we assumed a flat network topology. While this already yields precise estimates, Piper can be extended to handle hierarchical network topologies (see Appendix E).

## 6 Conclusions and Future Work

In this work, we presented a two-level dynamic programming approach that determines how to partition a provided model over a given number of accelerators, while optimally composing tensor and pipeline model parallelism with data parallelism and memory-saving optimizations. In our evaluation, we found that Piper finds high-throughput parallelization strategies that in many cases improve upon planners from prior work, and that Piper's power of finding non-trivial combinations of parallelism modes is especially beneficial in large-scale or memory-constrained scenarios.

While we consider pipeline parallelism schedules without pipeline flushes in this paper, one way of extending Piper to consider schemes with pipeline flushes is by adding the pipeline depth to Piper's dynamic programming state. However, this would worsen Piper's time complexity. We leave such extensions and improvements to the algorithm for future work. The main open question resulting from this work is automatically finding good tensor-parallelism schemes for each individual layer type (rather than the entire DNN operator graph). Piper can then determine how to best compose these per-layer tensor-parallelism schemes to obtain a high-performance training configuration for the entire model.

## Acknowledgments and Disclosure of Funding

This work was done in the context of Project Fiddle at MSR. During this work, the second author was at Stanford University, where he was supported in part by NSF Graduate Research Fellowship grant DGE-1656518. Any opinions, findings, and conclusions or recommendations expressed in this material are those of the authors alone.

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
