| | d | s | s/d | | | | | | | | | | | | | | | | | | | | | | | | | | |
|---|---|---|---|---|---|---|---|---|---|---|---|---|---|---|---|---|---|---|---|---|---|---|---|---|---|---|---|---|---|
| Stage 1, device 1 | 3 | 6 | 2 | 1 | 1 | 1 | | | | 4 | 4 | 4 | 1 | 1 | 1 | 7 | 7 | 7 | 4 | 4 | 4 | 10 | 10 | 10 | 7 | 7 | 7 |
| Stage 1, device 2 | 3 | 6 | 2 | | | 2 | 2 | 2 | | | 5 | 5 | 5 | 2 | 2 | 2 | 8 | 8 | 8 | 5 | 5 | 5 | 11 | 11 | 11 | 8 | |
| Stage 1, device 3 | 3 | 6 | 2 | | | | 3 | 3 | 3 | | | 6 | 6 | 6 | 3 | 3 | 3 | 9 | 9 | 9 | 6 | 6 | 6 | 12 | 12 | | |
| Stage 2, device 4 | 1 | 3 | 3 | | | | 1 | | 2 | | 3 | 1 | 4 | 2 | 5 | 3 | 6 | 4 | 7 | 5 | 8 | 6 | 9 | 7 | 10 | 8 | 11 |
| Stage 3, device 5 | 1 | 2 | 2 | | | | | 1 | | 2 | 1 | 3 | 2 | 4 | 3 | 5 | 4 | 6 | 5 | 7 | 6 | 8 | 7 | 9 | 8 | 10 | 9 |
| Stage 4, device 6 | 1 | 1 | 1 | | | | | | 1 | 1 | 2 | 2 | 3 | 3 | 4 | 4 | 5 | 5 | 6 | 6 | 7 | 7 | 8 | 8 | 9 | 9 | 10 |

Figure 5: Example of a pipelined schedule (in particular, PipeDream-2BW's schedule), with non-equal data-parallel degrees for different stages. Stage 1 has $d_1 = 3$ and uses devices numbered 1–3; stages 2–4 have $d_2 = d_3 = d_4 = 1$ and use devices numbered 4–6 respectively. Tensor parallelism is not used in this example. The suffix sums of data-parallel degrees ($s$) are shown in the figure. The forward pass, marked in blue, is assumed to take the same time as the backward pass, marked in green. The runtime of a single microbatch in stage 1 is assumed to be 3 times larger than the runtime in each of the stages 2, 3, and 4 (which makes for a balanced schedule since stage 1 has a data-parallel degree of 3). The number $\lceil s/d \rceil$ is the number of in-flight microbatches of every device in steady state. Stashed activations for these microbatches must be kept between their arrival (blue) and departure (green). For example, device 2 maintains stashed activations for microbatch sets $\{2\}, \{2, 5\}, \{5\}, \{5, 8\}, \{8\}, \{8, 11\}$ and $\{11\}$ at different points in time.

## A    Batch size and memory usage considerations

**Batch size considerations.**    Usually, a maximum safe batch size $B'$ (verified to not compromise statistical accuracy and model convergence) is known; for large modern language models, this could be on the order of 1024–2048 [20]. Our search space will consist of schedules with given batch size $B'$.

A batch consists of many microbatches; the total batch size is $B' = N \cdot b'$, where $b'$ is the microbatch size. The value of $b'$ should be chosen before Piper is run. If multiple choices of $b'$ are being considered, then we envision that single-layer runtimes would be profiled for multiple settings of $b'$ and then Piper is run for each of them (setting $N := B'/b'$). The returned TPS (Time-Per-Sample) should be more precisely understood as Time-Per-Microbatch, so it should be normalized by dividing by $b'$. Then, the lowest value (and the corresponding microbatch size) would be selected.

When a pipeline is in steady state, the number of microbatches "in flight" and being processed by some worker in the pipeline is equal to the sum $s = \sum_i d_i$ of data-parallel degrees of all stages. This is because with data parallelism, multiple devices are processing disjoint microbatches (devices process the same microbatch, possibly disjoint shards of it, with tensor parallelism).

In the pipelining schemes that we consider [17, 18], the number of microbatches in a batch ($N$) is at least $s$ (so that the pipeline is saturated). Furthermore, we must have the data-parallel degree of every stage be a divisor of $N$, so that the microbatches can be evenly split among data-parallel replicas.

Without tensor parallelism, $s$ is always equal to the number of devices used. Therefore, if more than $N$ devices are available, tensor parallelism (and Piper's fine-grained control over the sum $s$ of data-parallel degrees) is required to use all devices. In particular, even if $b' = 1$, tensor parallelism is required to use more than $B'$ devices.

**Memory usage.**    Consider a stage $j$ (out of $\ell$ total stages). In steady state, the number of micro-batches being processed by stages $j, ..., \ell$ is equal to $\sum_{i=j,...,\ell} d_i$ (in other words, $s$ from Piper's DP table for stage $j$). For each in-flight microbatch, we need to store all activations (if using PipeDream-2BW [18] and not using activation recomputation) or just input activations (if using PipeDream-2BW [18] and using activation recomputation) or stashed weights and activations (if using PipeDream [17]). If stage $j$ has data-parallel degree $d_j$, then these activations are partitioned equally among the devices assigned to this stage, which results in each of these devices needing to store $\lceil s/d_j \rceil$ such activation/weight stashes. See Fig. 5 for an example.

Besides this, each stage has other memory costs that do not depend on the number of in-flight microbatches (such as optimizer state, temporary buffers, or two sets of weights if using PipeDream-

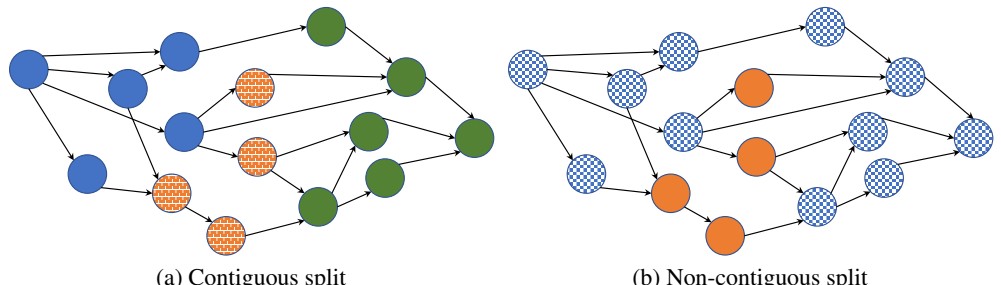

(a) Contiguous split           (b) Non-contiguous split

Figure 6: (a) Contiguous and (b) non-contiguous splits. Note that the brick-patterned orange nodes in (a) form a contiguous subgraph despite not being connected, and checked blue nodes in (b) form a non-contiguous subgraph despite being connected. Figure borrowed from Tarnawski et al. [23].

2BW [18]). Thus, for all these modes of execution, the memory usage can be modeled as

$$\sum_{v \in S} a_v \cdot \lceil s/d_j \rceil + \sum_{v \in S} b_v \,,$$

where $a_v$ and $b_v$ are some coefficients that are functions of the node/layer $v$ and of the TMPC selected for this node. (For instance, if some tensor parallelism scheme partitions the activations among devices, $a_v$ will decrease; if it [also] partitions the weights, $b_v$ will [also] decrease, and so on.)

## B  Data flow in tensor-parallelized stages

In this section, we discuss our model for the *edge-related* communication costs in the presence of tensor parallelism. Consider a subgraph/stage that is being tensor-parallelized. For each layer $v \in S$, there are certainly communication costs between the $t$ devices that are incurred in the course of the actual execution of $v$; these costs are modeled in the Time-Per-Sample quantity $X(v).p$ (whose value is obtained via profiling) and this section is not concerned with them. Rather, we are interested in modeling the transfers of tensors that correspond to edges. For simplicity let us concentrate on the forward pass; the backward pass is symmetric.

Let subgraph/stage $S \subseteq V$ be tensor-parallelized on $t > 1$ devices; for each $v \in S$, let $X(v)$ be the TMPC selected for that node/layer.

**Layers finish with full output activations on each device.**  Let $v \in S$. We will assume (and require from the schemes that the TMPCs describe) that, once $v$ is done executing, *each of the $t$ devices will hold the same, full, output activations*. This may necessitate adding a final synchronization step between the $t$ devices to the tensor parallelization configuration if, for instance, each device were to end up computing a $1/t$ part of the final output activation, partitioned along some dimension; now these need to be exchanged so that every device holds the entire output activation.

We believe that this assumption is natural, as most tensor parallelization strategies in practice already have this property; in particular, this is true for the Megatron-LM strategy [22] that we use in our evaluations. Furthermore, the assumption allows us to guarantee that our cost model is simple and tractable. Indeed, an alternative might be to allow strategies that leave $1/t$ of the output tensor on each device, which would give benefits in communication if the subsequent layer is also tensor-parallelized among $t$ devices and the first operation in that layer is partitioned in such a way that it could take $1/t$ of the input activation on each device. However, even then, we would not know whether the last operation in the previous layer and the first operation in the next one would have inputs and outputs partitioned *along the same dimension*. In order to know this, we would need to maintain additional state (e.g., the shape of the input and output tensors of each Tensor Model Parallelism Configuration of each layer), and only select compatible configurations of TMPCs in a stage, which

would be computationally costly and would require replacing the efficient heuristic for the knapsack subproblem with a more involved procedure.[3]

**Data flow over edges.** Now, consider an edge incoming to $v$ from some other node $u$. Which devices should send outputs to the $t$ devices that hold $v$? There are optimization/scheduling-type decisions to be made here. If, for instance, $u$ is also parallelized on some number $k \geq t$ devices, then it would make sense to send activations in a matching-like manner (e.g. if $k = t = 3$ and $u$ is processed on devices 1–3 while $v$ is processed on devices 4–6, then one could communicate activations like $1 \to 4, 2 \to 5, 3 \to 6$). We can certainly do this in the runtime; however, to make the problem structure tractable for dynamic programming, we proceed somewhat differently in the cost model. There are two cases depending on whether $u$ is also in the same subgraph $S$ as $v$, or not:

- **Case $u \notin S$:** We cannot afford to know (keep in the dynamic programming state) information about how many devices the subgraph where $u$ lies is partitioned on. Thus we must assume the worst case, which is that only one device holds the outputs of $u$. Now, we could have this single device send the outputs to each of the $t$ devices of $v$, but this would be difficult to account for when computing the cost of $u$'s subgraph, as we cannot [afford to] know $t$ at that time. (Of course, if this is the optimal mode of communication, then we can use it in the runtime; this gain will just not be reflected in the cost model.) What we do instead is:
  - If each of the $t$ devices needs the entire input activation, then we will have that single $u$-device send the output to the first of the $t$ devices, and then let them distribute those activations among themselves, so that they begin the actual processing each having the entire input activation. The TMPC should account for this synchronization by means of a communication cost $c^{\mathsf{fw}}(u, v)$ (in bytes) for each incoming edge that each of the $t$ devices will pay.
  - An exception: if each device only needs a $1/t$ fraction of that activation (e.g., because the first operation is already parallelized along an input-activation-splitting dimension), then the $u$-device sends $1/t$ of that activation to each device; in total, it will have sent one activation, so our prior accounting for the $u$-device's communication cost (one output tensor sent) was correct. In this case, the TMPC can have $c^{\mathsf{fw}}(u, v) = 0$ (when proposing a tensor parallelization where this input activation is only needed in shards).
  - For the special case of $t = 1$ (non-tensor-parallelized execution), we have $c^{\mathsf{fw}}(u, v) = 0$ as there are no communication costs in this case.
- **Case $u \in S$:** In this case, $u$ is tensor-partitioned among *the same* $t$ devices as $v$. Recall that we assumed above that after $u$ is done executing, each of these devices holds the entire output activation. Thus, in this case, nothing needs to be sent. (The TMPC will contain some $c^{\mathsf{fw}}(u, v)$ value, as it does not "know" whether $u \in S$ or not, but the algorithm will not use this value.)

For the backward pass, we proceed in the same way, but we call the relevant quantity $c^{\mathsf{bw}}(v, w)$, and it pertains to the *outgoing* edges of $v$. Note that in the backward pass, the communication on this edge goes in the reverse direction $w \to v$. That is, $c^{\mathsf{bw}}(v, w)$ is the additional communication cost that needs to be paid by $v$ after having received the backward tensor from $w$ on one device, in order to have this tensor on each device.

## C   Optimality gap of the knapsack heuristic

Let us remark that the knapsack heuristic is the only non-exact component of Piper. That is, if the heuristic did in fact yield always optimal solutions, then Piper would be an exact algorithm (that is, one finding optimal solutions) for the problem of finding the lowest-TPS global configuration in Piper's search space, subject to our cost model. Note that this is a sufficient condition, but far from necessary; for example, it would be enough that the knapsack heuristic be exact on the $O(|V|)$ instances $(S, t, d, s)$ that are part of some optimal global solution. Then, since the heuristic cannot

---

[3]Of course, if it turns out to be the case that two consecutive layers can be parallelized in such an advantageous way, then the ML compiler can still make a local optimization to perform communication more efficiently in the runtime; we simply do not model such possible gains in our cost model.

*under*estimate the TPS of any (other) configuration, Piper would find that solution (or some other optimal solution).

Moreover, we investigated whether the knapsack heuristic does in fact return suboptimal solutions in a practical workload. To this end, we extracted a random subset of around 30000 knapsack subproblem instances that arose when running Piper on our BERT-32 workload from Section 5 (8 GB RAM per GPU, $K = 512$ devices, batch size $N = 1920$). We then solved them to optimality using the Gurobi 8.1 [9] Integer Programming solver. We have determined that the heuristic indeed found the optimal solution in all instances.[4]

Finally, with minor modifications, one could also prove theoretical approximation guarantees for the heuristic (similar to Csirik et al. [5], for example).

## D  Running time analysis

Let us denote by $P$ the number of $t$-values for which there are TMPCs in the input; that is, $P := |\{t = 1, ..., K : (\exists v \in V)\, T(v, t) \neq \emptyset\}|$. Next, let us denote by $T$ the maximum number of TMPCs for one $(v, t)$ pair: $T := \max_{v,t} |T(v, t)|$.

We can implement Piper as follows. An outer loop iterates over downsets $D$ and then over sub-downsets $D'$: this gives $O(\mathcal{D}^2)$ iterations, where $\mathcal{D}$ is the number of downsets in $G$. Fix $D$ and $D'$ and define $S := D \setminus D'$.

We will first compute all $\mathrm{TPS}(S, t, d, s)$ values (for all $t$, $d$, $s$), and then use these values to update the $dp$ table. We first focus on the former.

We start by precomputing, for all $v \in S$, $t$ and TMPC $X \in T(v, t)$, the contribution of $X$ (if selected) to TPS and to memory usage, the latter in the form of quantities $a$ and $b$ to be used in the memory usage formula (see Section 3.5). We take all communication (except all-reductions for weight gradients when using data parallelism) into account here (in particular, the communication over the boundary edges of $S$). Note that we can perform such a precomputation as both TPS and memory usage are additive over nodes $v \in S$. Note also that this precomputation is independent of parameters $s$ and $d$. For fixed $S$, this step takes $O(P \cdot T \cdot (|S| + |E|))$ time, as we look at every edge in $E$ at most twice (once for each endpoint that it has in $S$).

Then, we iterate over all degrees $t$ and $d$ of tensor and data parallelism, respectively; as we are using $t \cdot d \leq K$ devices here, $d$ ranges from 1 to $\lfloor K/t \rfloor$. For each $(t, d)$, we need to solve one knapsack subproblem instance per $s$ (to compute $\mathrm{TPS}(S, t, d, s)$; recall that $s$ ranges from 1 to $N$). Here we can take advantage of the fact that memory usage only depends on $s$ through $\lceil s/d \rceil$, which means that it does not change within groups $s \in \{1, ..., d\}$, $s \in \{d + 1, ..., 2d\}$, $s \in \{2d + 1, ..., 3d\}$, and so on. Therefore there are only $\lceil N/d \rceil$ distinct knapsack subproblem instances that we need to solve. Finally, the knapsack heuristic can be implemented to run in time $O(|S|T^2 \log |S|)$. The total running time of this step (for fixed $S$) is thus

$$\sum_{t=1}^{P} \sum_{\substack{d=1 \\ d|N}}^{\lfloor K/t \rfloor} \lceil N/d \rceil \cdot |S|T^2 \log |S| \leq P \cdot |S|T^2 \log |S| \cdot \sum_{f|N} f \,,$$

where we substituted $f = N/d$. Thus we expect that Piper's running time will be smaller for "less composite" numbers $N$ such as powers of two; this can indeed be seen in Fig. 3. Asymptotically, we can bound $\sum_{f|N} f = O(N \log \log N)$ [10, Section 22.9]. Thus the running time of this step can be written as $O(P|S|T^2 N \log |S| \log \log N)$.

Finally we consider updating the $dp$ table. We loop $t$ (over some set of size at most $P$) and $d$ over divisors of $N$ from 1 to $K/t$ (as we must have $t \cdot d \leq K$). There are at most $P \cdot d(N)$ such pairs $(t, d)$, where $d(N)$ is the number of divisors of $N$. We also need to iterate over $k = t \cdot d, ..., K$ and $s = d, ..., N$ (as $t \cdot d \leq k$ and $d \leq s$), which gives $O(KN)$ pairs $(k, s)$. Then we perform the update of $dp[D][k][s]$ with $\max(dp[D'][k - t \cdot d][s - d], \mathrm{TPS}(D \setminus D', t, d, s))$, which takes constant time. Overall this step takes time $O(PKNd(N))$.

---

[4]We do note that in our evaluation, we always have $|T(v, t)| \leq 2$ for all $(v, t)$, and most layers $v$ are very similar (transformer layers). This might be making the knapsack subproblem easier. At the same time, we expect these properties to hold for many real-world workloads.

Taken together, we get a time complexity of

$$O\left(\mathcal{D}^2 \cdot P \cdot \left(T \cdot (|V| + |E|) + |V|T^2 N \log|V| \log\log N + KNd(N)\right)\right).$$

**Simplification.**   In practice, we expect $\mathcal{D} = O(|V|)$, $|E| = O(|V|)$, and $T, P = O(1)$. We also use the $\tilde{O}$-notation to suppress log factors. We then finally get a runtime of

$$\tilde{O}(|V|^2 N(|V| + K \cdot d(N))).$$

## E   Hierarchical network topologies

We have thus far assumed a network model where bandwidth constraints are placed on the incoming/outgoing communication of each device – that is, a flat network topology (one can think that these devices are all connected with an infinite-bandwidth switch). However, it is possible to extend Piper to a hierarchical model where we are given some $K_L$ devices ("superdevices") of level $L$, each of which consists of $K_{L-1}$ devices of level $L - 1$, and so on. The devices at each level $\ell = 1, ..., L$ (that are part of the same level-$(\ell + 1)$ device, if $\ell < L$) are interconnected with a network with bandwidth $B_\ell$. As an example, the infrastructure used for Fig. 4 was 8 NVidia DGX-A100 machines, interconnected (between machines) with a 25 GB/s Infiniband, and each containing 8 A100 GPUs, connected (inside the machine) with a 300 GB/s NVSwitch. The GPUs are level-1 devices, and DGX machines are level-2 devices. This corresponds to having $L = 2$, $K_1 = K_2 = 8$, $B_1 = 300$ GB/s and $B_2 = 25$ GB/s.

We refer to the dynamic programming algorithm from the main body of the paper as *single-level Piper*.

At a high level, the main technique needed to extend the dynamic program to multiple levels involves computing the optimal TPS for every number of devices *of some level*, not only for every downset, but for every contiguous set of nodes (that is, for every difference of two downsets).

The bottleneck of the system in such a hierarchical setting is either a level-1 device (whose load consists of compute and communication) or a higher-level device (whose load consists solely of communication).

**Basic version.**   Let us first ignore tensor parallelism, memory usage, and batch size. Without these, we are essentially in the setting of PipeDream's partitioning algorithm [17], albeit with an arbitrary DAG rather than a path-graph of layers. We compute values

$dp^\ell[D, D'][k] =$ best TPS (max-load) achievable by placing $D \setminus D'$ on $k$ devices of level $\ell$.

The final result will then be $\max_{k=1,...,K} dp^L[V, \emptyset][k]$.

There are two kinds of communication to account for:

- Communication over edges: note that an edge that crosses multiple levels needs to be accounted for at each level. One might worry that in the dynamic program, when considering a level $\ell$, a subgraph $D \setminus D'$ and an edge coming into it "from afar", we do not know (from the DP state) whether the other endpoint of that edge is placed on the same level-$\ell$ (or level-$(\ell + 1)$, level-$(\ell + 2)$, etc.) device as $D \setminus D'$. However, this is in fact not an issue: with this network model, we only need to account for the boundaries of the level-$\ell$ devices that we are currently forming, and we know that in this way, this edge will be accounted for at the boundary of each level that it actually crosses.

- All-reduce communication between data-parallel replicas: we can form stages that are replicated across multiple levels (e.g., level 1 and then again on level 2). We assume that in this case, the all-reduce implementation (say, a distributed parameter server approach) is cognizant of the hierarchical structure: for instance, the weights will be synchronized among layer-1 devices within the same level-2 device, and then among layer-2 devices (and so on, if there are more levels). Thus, here as well, at each level we are able to account for the data-parallel communication happening over that level's boundary. We also do not need to know how many devices are on the lower levels.

With this, we can write the DP recursion, in which we choose the boundary $D^\circ$ of the last stage[5] and its data-parallel degree $d$:

$$dp^\ell[D, D'][k] = \min_{\substack{D^\circ:\text{downset} \\ D' \subseteq D^\circ \subsetneq D}} \min_{\substack{d=1,\dots,k \\ d|N}} \max\Big(dp^\ell[D^\circ, D'][k-d],$$

$$\frac{1}{d}\left[\frac{4 \cdot \sum_{v \in D \setminus D^\circ} w_v \cdot \frac{d-1}{d} + 2 \cdot \sum_{(u,v) \in \delta(D \setminus D^\circ)} c(u,v)}{B_\ell}\right],$$

$$\frac{1}{d} \cdot dp^{\ell-1}[D, D^\circ][K_{\ell-1}]\Big) \ .$$

Here:

- $w_v$ is the size of weights associated with node $v$ (in single-level Piper, this information is kept in TMPCs, but we are ignoring tensor parallelism so far).

- $\delta(S)$ is the set of all edges crossing $S$.

- For $D^\circ = D'$, we have $dp^\ell[D^\circ, D'][k] = 0$ (for any $k$, $d$) as we are partitioning an empty set $D^\circ \setminus D' = \emptyset$ of nodes.

- The recursion arises as follows. We are using $d$ level-$\ell$ devices, in a data-parallel fashion, for the node set $D \setminus D^\circ$, and $k - d$ level-$\ell$ devices (recursively) for the remaining node set $D^\circ \setminus D'$. Then, the bottleneck (device attaining the maximum load) could be either

  - one of the $k - d$ level-$\ell$ devices or one of the lower-level devices inside them (this is the first $dp^\ell$ term),

  - one of the $d$ level-$\ell$ devices; recall from the above discussion that for these, we only look at communication, which is of two kinds: the first term corresponds to data parallel resync costs and the second term corresponds to communication over edges (here we account for the cost of bringing these activations one level down),

  - one of the lower-level devices inside the $d$ level-$\ell$ devices (this is the $dp^{\ell-1}$ term); we recursively partition $D \setminus D^\circ$ on all $M_{\ell-1}$ many level-$(\ell-1)$ devices inside each level-$\ell$ device, and we account for the data-parallelism speedup with the $\frac{1}{d}$ term.

- If $\ell = 1$, then instead of the last $dp^{\ell-1}$ term, we should account for compute costs, as in single-level Piper. We take the compute costs as $\sum_{v \in D \setminus D^\circ} p_v$, where $p_v$ is the latency of node $v$ (in single-level Piper, this information is kept in TMPCs), and we use the sum of this with the communication-related (second) term.

- One inaccuracy here is that we enforce $d \mid N$ on each level, but this does not necessarily guarantee that the final data-parallel degree for some level-$\ell$ stage (which is product of data-parallel degrees over all levels) will be divisible by $N$. If this is a concern, then one would need to additionally parametrize $dp^\ell$ by the number of microbatches in a batch ($N$) and make recursive calls with this quantity being divided by $d$ (i.e., instead of $dp^\ell$ depending on $dp^{\ell-1}$ in the DP recursion, we would have $dp^{\ell,N}$ depending on $dp^{\ell-1,N/d}$).

**Running time.** If we first precompute all sums of $w$, $c$ and $p$ values for all contiguous sets, which takes $O(\mathcal{D}^2 \cdot (|V| + |E|))$ time, then filling the $dp$ table on level $\ell$ requires computing $O(\mathcal{D}^2 K_\ell)$ values, each taking time $O(\mathcal{D}\min(K_\ell, d(N)))$, for a total time complexity of $O(\mathcal{D}^3 \sum_{\ell=1}^{L} K_\ell \min(K_\ell, d(N)))$. If $\mathcal{D}, |E| = O(|V|)$, then we get a total time of $O(|V|^3 K \min(K, d(N)))$.

**An improvement.** Note that, for the highest level, we in fact only need to compute $dp^L[D, D']$ for $D' = \emptyset$ (as we are using the same $D'$ for the recursive term $dp^L[D^\circ, D'][k-d]$). This can be very helpful if e.g. $L = 2$ and $K_2 \gg K_1$, as the time complexity will have terms $\mathcal{D}^2 K_2 d(N) + \mathcal{D}^3 K_1^2$ (rather than $\mathcal{D}^3 K_2 d(N)$).[6]

---

[5]We are forming a level-$\ell$ "superstage", which might be split further at lower levels.

[6]For this back-of-the-envelope calculation we assumed $K_2 > d(N) > K_1$.

**Crude memory accounting.**   We note that the above "basic version" can be augmented with an approximate memory accounting: namely, at level 1, we only consider forming a stage if its memory usage is at most $M$, and memory usage is counted as in Section 3.5, but we use $K$ as an upper bound for $s$. That is, for a subgraph $S$ and data-parallel degree $d$ we want to have

$$\left(\sum_{v \in S} a_v\right) \cdot \lceil K/d \rceil + \sum_{v \in S} b_v \leq M \,,$$

where $a$, $b$ are the memory usage quantities that in single-level Piper are stored in TMPCs. This approach is somewhat akin to ignoring the fact that later stages have lower memory usage since they need to store fewer activations.

**Precise memory accounting and batch size considerations.**   Now we would like to extend the above "basic version". Recall that in single-level Piper, these are controlled by a parameter $s$, which is a suffix sum of data-parallel degrees over stages. Let us remark that now, for a "superstage" that is replicated over more than one level, we should take the product of its data-parallel degrees.

We start with a rather heavy-handed extension of the above dynamic program to take the parameter $s$ into account. Consider, as previously, forming a new stage for the subgraph $D \setminus D^\circ$. To correctly estimate the memory usage of this (super)stage, we need to know the sum of data-parallel degrees from this stage to the last stage (not just the last stage of the split of $D \setminus D^\circ$, but the last stage of the split of $D$). Also, the recursive call for $D^\circ \setminus D'$ will need to know its similar sum of data-parallel degrees, but we would not know this unless we also control the sum of data-parallel degrees in the split of $D \setminus D^\circ$ itself. This means that unfortunately we need to use another parameter, which we will denote by $x$.

Namely, we can compute

$$dp^\ell[D, D'][k][s, x] = \text{best TPS (max-load) achievable by placing } D \setminus D' \text{ on } k \text{ devices of level } \ell$$
$$\text{assuming that the sum of data-parallel degrees in } D\text{'s split is } s$$
$$\text{and that the sum of data-parallel degrees in } (D \setminus D')\text{'s split is } x.$$

Now, in the DP recursion, we will also select a sum $x^\circ$ of data-parallel degrees for the new stage (for $D \setminus D^\circ$). Note that $x^\circ$ must be divisible by $d$. Then, the sum of degrees for $(D^\circ \setminus D')$'s split will become $x - x^\circ$, and the sum of degrees for $D^\circ$'s split will become $s - x^\circ$. The lower-level split (of $D \setminus D^\circ$) needs to have sum of degrees $x^\circ/d$. Moreover, since this is replicated $d$-wise, the effect of the $s$ microbatches waiting in the pipeline on memory usage can be obtained by dividing by $d$ (taking the ceiling; similarly as in Appendix A). Let us write the full formula:

$$dp^\ell[D, D'][k][s, x] = \min_{\substack{D^\circ:\, \text{downset} \\ D' \subseteq D^\circ \subsetneq D}} \min_{\substack{d=1,\ldots,k \\ d \mid N}} \min_{\substack{x^\circ = d, 2d, \ldots \\ x^\circ \leq \min(x, d \cdot K_{\ell-1})}} \max \Big( dp^\ell[D^\circ, D'][k-d][s - x^\circ, x - x^\circ],$$
$$\frac{1}{d}\left\lceil \frac{4 \cdot \sum\limits_{v \in D \setminus D^\circ} w_v \cdot \frac{d-1}{d} + 2 \cdot \sum\limits_{(u,v) \in \delta(D \setminus D^\circ)} c(u,v)}{B_\ell} \right\rceil,$$
$$\frac{1}{d} \cdot dp^{\ell-1}[D, D^\circ][K_{\ell-1}][\lceil s/d \rceil, x^\circ/d] \Big) \,.$$

As previously, we adjust the recursive relations for $\ell = 1$ appropriately (using the parameter $s$ to account for memory usage, as in single-level Piper, and ensuring that the parameter $x$ is equal to the sum of data-parallel degrees of the split).

**Improvements.**   The proliferation of parameters leads to a running time that is still polynomial (as long as $\mathcal{D} = \text{poly}(|V|)$), but of a very high degree, and most likely no longer practical. However, we can again optimize for the typical scenario we are targeting, where $L = 2$ and $K_1$ is small. To that end, notice that for $\ell = L$:

- we still only need to consider $D' = \emptyset$,

- and then, the parameter $x$ is unnecessary, as in this case, $D \setminus D' = D$ and so we must have $x = s$.

If $L = 2$, then only level $\ell = 1$ remains, and if $K_1$ is small (e.g., $K_1 = 8$), the above dynamic program should be feasible to run.

We note that another possible, albeit lossy, optimization is to upper-bound $s$ by $s \leq N$ whenever counting memory usage, similar to the approach proposed above for "crude memory accounting"; then the parameter $s$ can be removed, but we still control the sum of data-parallel degrees with the parameter $x \leq N$.[7]

**Tensor parallelism.** Finally, we extend the above with tensor parallelism. Following recent work [18, 19], we focus on tensor parallelization within a single level-2 device (e.g., an NVidia DGX machine), as very fast communication is necessary for tensor parallelization techniques to be efficient. This means that we only need to consider tensor parallelism for $\ell = 1$, and thus level 1 essentially becomes single-level Piper, but

- extended by computing the best configuration for every contiguous set $D \setminus D'$ (rather than every downset $D$), as above,
- extended by including the parameter $x$, as above,
- but executed for a smaller number $K_1 \ll K$ of devices.

# F   Further evaluation results

In Table 1, we present a comparison of Piper to the **equi** baseline for two other DNN models. We do not use tensor parallelism for these experiments. These results demonstrate that equi-partitioning often fails to recover high-throughput partitionings when compared to more involved (e.g., dynamic programming based) partitioning algorithms such as Piper or other recent work [17, 23].

---

[7]Of course, there is no reason to control the sum of data-parallel degrees unless we are using tensor parallelism, since absent tensor parallelism, this sum is always equal to the total number of devices $K = \prod_{\ell=1}^{L} K_\ell$.

| Number of devices ($K$) | Memory per device (GB) | Piper (TPS) | equi (TPS) | equi normalized to Piper (throughput) | Piper's runtime |
|---|---|---|---|---|---|
| GNMT | | | | | |
| 2 | 2.5 | 263.354 | 325.026 | 0.810× | 290.2s |
| 2 | 3.5 | 260.227 | 260.227 | 1.000× | 288.9s |
| 4 | 1.2 | 137.131 | OOM | 0.000× | 387.2s |
| 4 | 2.5 | 131.677 | 162.513 | 0.810× | 408.5s |
| 8 | 1.2 | 68.566 | 137.518 | 0.499× | 572.8s |
| 8 | 2.4 | 65.839 | 81.257 | 0.810× | 572.3s |
| 16 | 1.2 | 34.263 | 68.759 | 0.498× | 792.7s |
| 32 | 0.8 | 17.504 | 71.188 | 0.246× | 1081.2s |
| Resnet50 | | | | | |
| 8 | 4.0 | 73.493 | OOM | 0.000× | 4.7s |
| 8 | 8.0 | 63.434 | 164.159 | 0.386× | 4.9s |
| 8 | 16.0 | 58.558 | 108.885 | 0.538× | 5.2s |
| 16 | 2.0 | 76.521 | OOM | 0.000× | 7.4s |
| 16 | 4.0 | 35.333 | OOM | 0.000× | 7.3s |
| 16 | 8.0 | 31.634 | 82.080 | 0.385× | 8.0s |
| 16 | 16.0 | 29.279 | 54.443 | 0.538× | 8.9s |
| 32 | 1.5 | 50.977 | OOM | 0.000× | 9.6s |
| 32 | 2.0 | 24.626 | OOM | 0.000× | 10.0s |
| 32 | 4.0 | 17.624 | OOM | 0.000× | 10.3s |
| 32 | 8.0 | 15.758 | 41.040 | 0.384× | 10.8s |
| 32 | 16.0 | 14.639 | 27.221 | 0.538× | 12.5s |
| 64 | 1.5 | 16.992 | OOM | 0.000× | 13.0s |
| 64 | 2.0 | 11.334 | OOM | 0.000× | 13.0s |
| 64 | 4.0 | 8.667 | OOM | 0.000× | 13.5s |
| 64 | 8.0 | 7.879 | 20.520 | 0.384× | 15.2s |
| 64 | 16.0 | 7.320 | 13.611 | 0.538× | 19.1s |
| 128 | 1.0 | 19.856 | OOM | 0.000× | 17.3s |
| 128 | 1.5 | 7.025 | OOM | 0.000× | 17.4s |
| 128 | 2.0 | 5.406 | OOM | 0.000× | 18.2s |
| 128 | 4.0 | 4.306 | OOM | 0.000× | 20.0s |
| 128 | 8.0 | 3.895 | 10.260 | 0.380× | 21.7s |
| 128 | 16.0 | 3.660 | 6.805 | 0.538× | 27.6s |

Table 1: Results of comparing Piper to the baseline **equi** on two further DNN models, GNMT and Resnet50, without tensor parallelism. We use profiles (on an nVidia GTX 1080Ti GPU) from prior work [23]. For both, we use multiple settings for the number of devices and the memory limit. For GNMT, the throughput of **equi** was on average 67% of that of Piper, and in one scenario **equi** could not find any memory-feasible solution while Piper was able to. For Resnet50, the throughput of **equi** was on average 46% of that of Piper, and we found more scenarios where **equi** could not find any feasible solution.