# OpenReview forum: "Piper: Multidimensional Planner for DNN Parallelization"
_NeurIPS.cc/2021/Conference — NeurIPS 2021 Poster_

### Official Review · Reviewer_bfSz · 2021-07-05

**Rating:** 7
**Confidence:** 4

**Summary:**

Parallel/distributed training of machine learning models is a big and complex task. This paper proposed a dynamic programming algorithm in searching optimal parallelization configurations for machine learning models, considering tensor parallelization, data parallelization, and pipeline parallelization. Designing such algorithm is important and interesting, and potentially relieve machine learning app writers from manually designing parallelization configurations. To provided an optimal (or close to optimal) parallelization configuration without breaching memory limitation is already good contributions in real-world scenarios.

**Limitations And Societal Impact:**

The authors didn't say that their work can potentially solve the climate change problem, but that is totally fine.

**Main Review:**

Finding parallelization configurations is not a new problem, but the problem is so complex that new contributions are always needed. This paper didn't bring up any new ways to parallelize machine learning models, but tried to provide a better (or optimal) combinations of known parallelization techniques, which is also very interesting. Finding good combinations of parallelization techniques isn't new idea either, but previous publications have missed opportunities.

Describing such dynamic programming algorithm is not easy. This paper did a good job in clearly presenting the algorithm with formal definitions. The main dynamic programming recursion is deciding where to cut the next pipeline stage from the difference of 2 downsets. This is a clearly good idea. Although the algorithm can probably be optimized to not consider very small pipeline stages or very large pipeline stages, since all the pipeline stages should still probably have similar latencies for optimal performance, this doesn't hurt the correctness of the algorithm.

First question I have is: are the authors only considering PipeDream-style pipelining, and discarding GPipe-style pipelining? It is unclear to me if the algorithm is considering the extra cost of pipeline-filling and pipeline exhausting in GPipe. If the answer is yes, that is probably fine, since PipeDream-style pipelining is probably higher in throughput. If the answer is no, I would like to hear why.

Each pipeline stage is handling a part of the machine learning model, including both its forward computations and backward computations. This is probably a reasonable limitation. Within each pipeline stage, the computation of TPS depends on a heuristics. This is also quite reasonable I think, since solving this problem exactly is almost impossible, not only because the problem is NP-hard, but also because the estimations of memory consumption is very unlikely to be accurate. Each device might reserve memories for other purposes, and some details of memory recycling can be chaotic. Using a heuristics and trying not to use to the memory limit is a wise choice.

However, I think one significant problem of this paper is that the handling of activation recomputation depends on the TMPC. That is to say, recomputation can only happen in a layer. For instance, given a pipeline stage of A -> B -> C -> D -> E, the backward computation of the layers (A', B', C', D', and E') might choose to recompute something in each layer, but you can never recompute multiple layers together, such that the backward computation becomes B -> C -> D -> E' -> D' -> C' -> B' -> A'. In my example, the whole layers of B C and D are recomputed. This limitation might not be so significant in this paper, since the evaluations used models that have huge layers (such as transformer layers). But if the model has many small layers, then it is more likely that consecutive layers need to be recomputed to save memory. How difficult is it for this type of recomputation to be considered?

When modeling the cost of parallelization, the cost of communication is for sure critical. That is why when the authors say that in tensor parallelization, each of the t devices has a full copy of the inputs and outputs, the reviewer is a bit doubtful. One likely reason for this to not be a problem, is that tensor parallelization is not heavily used or optimized at runtime, or most tensor parallelizations likely require full copy of the inputs and outputs to be on all t devices. Based on the supplemental data, it seems that the latter is the reason? This limitation is thus likely reasonable, since the modeling of cost doesn't have to be very very accurate.

In the algorithm, it seems that each pipeline stage has one d (data parallelization) and t (tensor parallelization) value. This is a limitation such that layers in the same pipeline stage must have the same d and t. The reviewer is not very sure how big this limitation is. For instance, if a pipeline stage has layer A -> B -> C -> D, and only layer C has a very efficient tensor parallelization of t = 4, does that mean that the algorithm cannot make use of it, since the other layers (A, B, and D) do not have a TMPC for t = 4? I guess the authors can create a (TMPC of t = 4) by duplicating a (TMPC of t = 1) 4 times (just like data parallelization)? but I am not sure if that works, since it breaks the conceptional separation of data parallelization and tensor parallelization. The authors can certainly argue that, the lack of TMPCs is not the problem for this paper. However, this limitation seems to be quite practical. Can the authors elaborate here?

Of course, in the end, we care about if the new algorithm is providing better throughput than existing methods, and if we learn anything exciting from the dynamic searching algorithm. Evaluation seems to be the main reason of rejection from previous submissions. Indeed, it sticks out that most evaluations are just running the algorithm itself and presenting the estimated throughput. The Figure 1 showed interesting ablation tests, and demonstrated that removing some options in the search space caused worse throughput. It does increase the confidence that the algorithm is considering the right things, but it makes reviewers less confident, since the modeling/estimation itself could be incorrect. The added Figure 4 does boost that confidence.

The take aways are not so exciting but rather expected. The "Power of Piper's search space" is particularly interesting, since it provides a theoretical explanation of why Piper is better.

In summary, this paper addressed a very relevant problem of high importance. The solution is a small delta to what has been published, but it is nevertheless important. There are 2 significant limitations (about recomputation and fixed d/t for all layers in one stage), which the reviewer hope to get elaborations from the authors. The evaluations are not ideal, but the challenges of running all configurations in real machines is probably too high to request.

**Time Spent Reviewing:**

15 hours

---

> ### Author Response · Authors · 2021-08-10
> **Response to Reviewer bfSz**
>
> Thank you for your very detailed and insightful review! Please find our responses below.
>
>
> - *Although the algorithm can probably be optimized to not consider very small pipeline stages or very large pipeline stages, since all the pipeline stages should still probably have similar latencies for optimal performance, this doesn't hurt the correctness of the algorithm.*
>
> That is a very good idea to optimize the efficiency of the dynamic program. In a similar vein, for example, if we already know a solution with Time-Per-Sample X, then we can cut off any computation branch that would necessarily result in a Time-Per-Sample larger than X. (Our proof-of-concept implementation does not have such optimizations.)
>
>
> - *are the authors only considering PipeDream-style pipelining, and discarding GPipe-style pipelining?*
>
> As written, Piper can handle PipeDream and PipeDream-2BW schedules. However, it can also be made to handle GPipe-style pipelining (with periodic flushes) by additionally incorporating the pipeline length (the number of stages) into the dynamic program state. Also, you are right that PipeDream or PipeDream-2BW schedules can achieve higher throughput compared to GPipe and this is the reason we consider them as the default scheme in Piper.
>
>
> - *Each pipeline stage is handling a part of the machine learning model, including both its forward computations and backward computations. This is probably a reasonable limitation.*
>
> Yes: if we are able to fit the model across all accelerators, then it is better to avoid having to communicate weights between devices (which would be necessary if the forward and the backward pass operators were placed on different devices).
>
>
> - *How difficult is it for this type of recomputation to be considered?*
>
> This is a great question, and something that we also thought about. In fact, Piper can handle this mode of Activation Recomputation (AR), as follows. We mark TMPCs corresponding to AR (with a boolean flag). Recall that for every edge (u,v) we know the size c(u,v) of the tensor communicated over it. Therefore, if a node v uses AR, then for its every incoming edge (u,v), if u is in the same stage and also uses AR, we can account for the fact that v does not need to store these incoming tensors for every in-flight microbatch by essentially subtracting c(u,v) from the memory-usage coefficient `a` of the AR-enabled TMPC for v. (This can be done in the knapsack heuristic.)
>
>
> - *when the authors say that in tensor parallelization, each of the t devices has a full copy of the inputs and outputs, the reviewer is a bit doubtful. One likely reason for this to not be a problem, is that tensor parallelization is not heavily used or optimized at runtime, or most tensor parallelizations likely require full copy of the inputs and outputs to be on all t devices. Based on the supplemental data, it seems that the latter is the reason? This limitation is thus likely reasonable*
>
> Indeed, all tensor-parallelism schemes that we know of end with all outputs on every device, in particular the Megatron-LM scheme for Transformer layers that we use in the evaluations. Usually, for example, layers end with a dropout, which requires an all-reduce anyway (as it contains a batch normalization). However, note also that if a scheme only requires input activation in shards, then we can handle this without accounting any spurious communication costs (see Appendix B, lines 577-582).
>
>
> - *This is a limitation such that layers in the same pipeline stage must have the same d and t. The reviewer is not very sure how big this limitation is. (...)*
>
> To some degree, these limitations are not only design decisions made for the sake of simplicity or tractability, but also aligned with the capabilities of pipeline execution systems / ML compilers that Piper is targeting; even having **different** stages with different data-parallelism and tensor-parallelism degrees is currently considered as difficult to implement and missing from most frameworks (let alone having different degrees inside a single stage). (The results of this paper will hopefully convince people that it is worth implementing this.)
>
> However, what you are suggesting is very reasonable. We did not need to do this in the evaluation since all nodes had TMPCs for the same degrees t, but yes - one can simply duplicate a TMPC for t=1 and use it for any t. This would correspond to having that layer on a single device, with the t-1=3 others being idle while this layer is processing. If you are instead suggesting to have all 4 devices work in a data-parallel fashion for that single layer - that is, essentially, having multiple dissimilar stages colocated on the same set of devices - then it seems like a good idea to extract more performance, but no such pipelined schedules have been proposed yet, to the best of our knowledge (one has to figure out how to schedule the flow of samples, understand the memory usage etc.). We are quite confident that Piper can be extended to such execution schemes too, as it is a general framework.
>
>
> - *Indeed, it sticks out that most evaluations are just running the algorithm itself and presenting the estimated throughput.*
>
> We were fortunate to get access to a 64-GPU nVidia A100 system for the submission, which allowed us to empirically confirm and demonstrate that the predicted and measured throughputs very closely match. Given this close correspondence, we have concentrated on performing experiments using the predicted throughput. We believe that this is sound. We would like to also note that it is not practical for us to e.g. obtain access to a system with 2048 GPUs; and that the experimental setup from Figure 4 corresponds to the one for Figure 1a (for k=64). While in Figure 1a we do not plot the results for k=64, they look like an average of those for k=32 and k=128.
>
>
> - *The take aways are not so exciting but rather expected.*
>
> We suppose that this can be another factor that increases the confidence that the modeling is correct.
>
>
> - *The solution is a small delta to what has been published, but it is nevertheless important.*
>
> One further reason we would argue that it is not a small delta is that a major conceptual contribution of our work is the two-level perspective that we employ (and the notion of TMPCs). It reduces the extremely hard problem of finding good tensor-parallelism schemes for the entire DNN operator graph (in conjunction with the other modes of parallelism) to the much more tractable problem of finding just good tensor-parallelism schemes for single layers.
>
>
> - *The evaluations are not ideal, but the challenges of running all configurations in real machines is probably too high to request.*
>
> Thank you for understanding our limitations.  We had to bend over backwards for the submission to get access to a large cluster of 64 nVidia A100 GPUs - and we did this precisely to empirically evaluate the close correspondence of predicted and measured throughput at the largest scale we could get our hands on (even if it was for a short duration). The entire goal of showing such a correspondence was to bolster the confidence in the predicted throughputs we present in the rest of the paper.  Furthermore, such predictions do not require utilizing the large number of resources that would otherwise have been used for training -- thus being resource and energy footprint conscious for this work.

---

### Official Review · Reviewer_eU38 · 2021-07-12

**Rating:** 6
**Confidence:** 5

**Summary:**

This paper presents a search based method to plan DNN parallelism. The search space includes pipelining, recomputation choices and data partitioning.


**Main Review:**

First of all, this paper studies an important problem of automatic parallelism. Two of the primarily related works are FlexFlow and Pipedream. Piper's search space can be viewed as the combination of the above two, plus the choice of recomputation. The overall contribution over the prior work is slightly incremental.

Given the limited technical contributions, the paper would benefit from stronger empirical evaluations. On that end, it would be beneficial to show a wide variety of model settings and how Piper applies to those models. It would also be useful to bring baselines that reflect the search space used by FlexFlow and PipeDream. Finally, it would be helpful to evaluate Piper's plan on a real end-to-end setting and see the benefit brought by the additional configuration space.

The paper is borderline due to the above considerations.


**Time Spent Reviewing:**

1

---

> ### Author Response · Authors · 2021-08-10
> **Response to Reviewer eU38**
>
> Thank you for taking the time to review our paper! Please find our responses below.
>
>
> - *The overall contribution over the prior work is slightly incremental.*
>
> One point to which we would like to draw more attention is that a major conceptual contribution of our work is the two-level perspective that we employ (and the notion of TMPCs). It reduces the extremely hard problem of finding good tensor-parallelism schemes for the entire DNN operator graph (in conjunction with the other modes of parallelism) to the much more tractable problem of finding just good tensor-parallelism schemes for single layers.
>
> - *On that end, it would be beneficial to show a wide variety of model settings and how Piper applies to those models.*
>
> Thank you for this suggestion, which has prompted us to bolster our evaluation. Let us begin by remarking that our evaluations focused on huge Transformer-based language models such as BERT or GPT-3, as these form a large and very important class of models. It is this class that has seen exponential growth in recent years. In addition, their training is particularly expensive, which amplifies the benefits from optimal parallelization. However, for more branching large DAGs that might arise in the future, we would in fact expect to see even larger gains - versus baselines that e.g. do equi-partitioning of the DNN - than we see in our BERT evaluations.
>
> To obtain a more complete picture, during the rebuttal period we have managed to source data for two more models, GNMT and Resnet50, profiled on nVidia GTX 1080Ti GPU, and evaluate Piper on them. (Note that GNMT is a DNN with high branching.) For both, we use multiple settings for the number of devices and the memory limit. We are happy to add these results to the final version. As expected, Piper remains efficient and outperforms the baselines. Here let us report some numerical highlights from the comparison with the baseline `equi`, which corresponds to the planner from the prior work PipeDream-2BW (equi-partitioning layers among the stages, trying all possible numbers of stages, data-parallel degrees, and enabling or disabling activation recomputation). For GNMT, the throughput of `equi` was on average 68% of that of Piper, and in one scenario `equi` could not find any memory-feasible solution while Piper was able to. For Resnet50, the throughput of `equi` was on average 46% of that of Piper, and we found more scenarios where `equi` could not find any feasible solution.
>
>
>
> - *It would also be useful to bring baselines that reflect the search space used by FlexFlow and PipeDream.*
>
> Please note that:
> 1. the baseline "equi-no TP" is essentially the planner from PipeDream-2BW;
> 2. the planner from PipeDream would be "no TP and no AR" (that is, no better than the worse of the two baselines "no TP" and "no AR");
> 3. Tarnawski et al. (NeurIPS 2020) would be "no TP, no AR and no DP" (that is, no better than the worst of these three baselines);
> 4. FlexFlow does not consider pipelining, which is essential to even get any reasonable throughput when the model is much larger than the capacity of a single accelerator.
>
> (Here we ignore the fact that we are also accounting memory usage much more precisely than these prior works.)
>
>
> - *Finally, it would be helpful to evaluate Piper's plan on a real end-to-end setting and see the benefit brought by the additional configuration space.*
>
> We were fortunate to get access to a 64-GPU nVidia A100 system for the submission, which allowed us to empirically confirm and demonstrate that the predicted and measured throughputs very closely match. We did so at the largest scale we could get our hands on (even if it was for a short duration). Given this close correspondence, we have concentrated on performing experiments using the predicted throughput. We believe that this is sound. We would like to also note that it is not practical for us to e.g. obtain access to a system with 2048 GPUs; and that the experimental setup from Figure 4 corresponds to the one for Figure 1a (for k=64). While in Figure 1a we do not plot the results for k=64, they look like an average of those for k=32 and k=128.

---

> > ### Comment · Reviewer_eU38 · 2021-08-26
> > **comment**
> >
> > Thanks for updating the experiments and baselines. it would be great to incorporate these notes and relations in the final submission.

---

### Official Review · Reviewer_cpuf · 2021-07-16

**Rating:** 5
**Confidence:** 4

**Summary:**

This paper proposes Piper, an efficient dynamic programming algorithm to combine data parallelism, tensor model parallelism, pipeline model parallelism, and activation rematerialization.
The paper evaluates the algorithm on BERT models with real per-layer profiling data and simulated end-to-end results.
It provides insightful analyses of the results.

**Limitations And Societal Impact:**

The author adequately addressed the limitation of this paper in Section 6.

**Main Review:**

Originality and clarity:
This paper tackles an important problem. The DP algorithm and the idea of using TMPC are novel.
The paper is well written with the necessary background information. It precisely discusses its relations to prior pipeline-parallelism works.

Strength:
1. It is the first work that uses dynamic programming to search in a non-trivial search space that includes data parallelism, tensor model parallelism, pipeline model parallelism, and activation rematerialization.
2. Clear analysis and explanation of the results.

Weakness:
1. The evaluation results in Figure.1 are not measured in real systems. They are obtained by simulation. The y-axis
"throughput" is computed by formulas proposed by authors under various assumptions. It does not reflect the
behavior of real systems. As a paper targeting a system problem, only running evaluation in a simulator is not
enough. The authors show real measured throughput in Figure.4, meaning that the authors have access to the machines.
Could you please use real profiling data in Figure.1, at least for small-scale experiments? The insight drew from these simulation experiments can be misleading if we do not consider some important factors in real systems.
2. In the evaluation, the strategy generated by this paper only provides significant advantages in unrealistic settings. For example, it only shows significant advantages over "equi-no TP" on settings with an extremely low memory budget or an extremely high number of devices.
Training large models under an extremely low memory budget distributedly is not realistic and nobody will do this. When the number of devices becomes large, the simulator model used in the paper is not accurate anymore.
3. The paper does not consider the effect of batch size when measuring the throughput of TMPC. A small batch size typically cannot utilize the full compute power of a modern high-end GPU. Even with the same tensor model parallel configuration, different batch sizes can have different TPS.


Additional questions:
1. In Figure.2, the blue line (Piper) is almost a straight line. Does this mean Piper can get perfect **strong** linear scaling? I feel like this is impossible in a real system. For example, even in this paper (https://arxiv.org/pdf/2104.04473.pdf), they can only get good **weak** scaling in a cluster with very high internode bandwidth.


**Time Spent Reviewing:**

2

---

> ### Author Response · Authors · 2021-08-10
> **Response to Reviewer cpuf**
>
> Thank you for taking the time to review our paper! Please find our responses below.
>
>
> - *It provides insightful analyses of the results. / Clear analysis and explanation of the results.*
>
> We are gratified that you find the analysis of the results interesting.
>
>
> - *The authors show real measured throughput in Figure.4, meaning that the authors have access to the machines. Could you please use real profiling data in Figure.1, at least for small-scale experiments?*
>
> We were fortunate to get access to a 64-GPU nVidia A100 system for the submission, which allowed us to empirically confirm and demonstrate that the predicted and measured throughputs very closely match. Given this close correspondence, we have concentrated on performing experiments using the predicted throughput. We believe that this is sound. We would like to also note that it is not practical for us to e.g. obtain access to a system with 2048 GPUs; and that the experimental setup from Figure 4 corresponds to the one for Figure 1a (for k=64). While in Figure 1a we do not plot the results for k=64, they look like an average of those for k=32 and k=128.
>
>
> - *In the evaluation, the strategy generated by this paper only provides significant advantages in unrealistic settings.*
>
> We would like to remark that, in the perspective of training huge models on many accelerators, a few-percent advantage is already significant. For example, it is reported that training GPT-3 costs tens of millions of dollars (using thousands of accelerators for 3 or more months).
>
>
> - *Training large models under an extremely low memory budget distributedly is not realistic and nobody will do this.*
>
> It is true that people are not training large models on accelerators with 1GB or 2GB of memory (except possibly for some memory-constrained FPGAs and ASICs). However, we include these results to highlight the very realistic (if slightly futuristic) scenario of a huge DNN whose every layer is 40x the size of a single Transformer layer from our BERT32 model. Then e.g. Figure 1c models the task of training this DNN using 80GB accelerators (think of scaling both the layer memory usage and the accelerator memory capacity by the same factor, which yields an equivalent problem instance). Similarly, we use the setup of Figure 1d to model a scenario where even a single layer does not fit on an accelerator (80x the size of that single Transformer layer).
>
>
> - *When the number of devices becomes large, the simulator model used in the paper is not accurate anymore.*
>
> While it is true that network performance characteristics do exhibit variance on public cloud providers, in many deployments of private large AI supercomputers for such language models, cloud providers do advertise strong perf SLAs (such as a guaranteed full bisection bandwidth) - thus making the assumptions in the simulator reasonable even at such large scales. Nevertheless, we note your point: incorporating flaky networks in the model is an interesting avenue for future work for the community at large.
>
>
> - *The paper does not consider the effect of batch size when measuring the throughput of TMPC. A small batch size typically cannot utilize the full compute power of a modern high-end GPU. Even with the same tensor model parallel configuration, different batch sizes can have different TPS*
>
> In this answer we are assuming you mean the microbatch size; please kindly let us know if you are referring to something else (e.g. the minibatch size).
>
> Piper assumes a given fixed microbatch size, and the input TMPCs should reflect running times / memory usage / etc. for that microbatch size. (One should really think of Time-Per-Sample as Time-Per-Microbatch.) If the user wants to choose one of multiple possible microbatch sizes, they should build an outer loop over Piper, providing Piper with profiles (TMPCs) taken for each microbatch size (and dividing the optimal max-load by the microbatch size to normalize). See also Appendix A.
>
>
> - *In Figure.2, the blue line (Piper) is almost a straight line. Does this mean Piper can get perfect strong linear scaling? I feel like this is impossible in a real system. For example, even in this paper (https://arxiv.org/pdf/2104.04473.pdf), they can only get good weak scaling in a cluster with very high internode bandwidth.*
>
> Figure 2 does not show strong scaling; it shows weak scaling, because the batch size (more precisely, the number of microbatches in a minibatch) is allowed to be anything up to the given bound N (also see Appendix A for more details). Thus the effective batch size of the optimal configurations is growing as the number K of devices increases (up to the threshold K=N). Hence, these results are consistent with the cited paper, which also shows good weak scaling in a cluster.

---

### Official Review · Reviewer_BM6L · 2021-07-17

**Rating:** 6
**Confidence:** 4

**Summary:**

This paper proposes a novel algorithm to find a pipeline parallel configuration where each pipeline stage also includes data and tensor model parallelism. The method assumes that the input computational graph consists of multiple “layers” that form a DAG and each layer has a  set of predefined tensor model parallelism configurations. The algorithm is a two-stage dynamic programming algorithm and is mainly evaluated in a simulated environment.

**Limitations And Societal Impact:**

See weaknesses in the main review section.



**Main Review:**

I lean towards accepting this paper. Although there are limitations with the proposed method and the evaluation method, I believe the proposed two-step problem formulation and dynamic programming algorithm are novel and should be useful for the community. Detailed comments:

Strengths:
- The two-step formulation proposed for pipeline and data/tensor-model parallelism is novel and will greatly reduce the complexity of the overall search space. I believe this is a positive contribution and will be useful for future work working in space.
A two-stage dynamic based on the two-step formulation solves the problem.
- The paper is clear and easy to follow.

Weaknesses:
- The algorithm assumes the knowledge of predefined “layers” and for every layer, there is a list of predefined model-parallel methods. For a more complicated neural network, this is not an easy task. This limits the application of the method on more complicated neural networks.
- There are non-negligible limitations on the experiments in the paper:
  - The paper only evaluates the BERT model. In this case, the DAG formulation actually degenerates to a chain, making the algorithm proposed in the paper overkill. The method should be evaluated on more neural networks with more complex architecture (that actually forms a DAG).
  - Most of the evaluations in the paper are simulated. Although Figure 4 partly proves the simulation is realistic, I’m still not convinced and I would like to see larger-scale experiments.

Minor Issues:
- Some key concepts will be easier to be understood with examples (e.g. Downsets).
Line 26: Maybe also add \emph for “pipelined model parallelism”?

Other questions
- Do you profile the Time-Per-Sample for every downset? How long does this take? Will this be a potential bottleneck when the neural network structure getting more complex?



**Time Spent Reviewing:**

3

---

> ### Author Response · Authors · 2021-08-10
> **Response to Reviewer BM6L**
>
> Thank you for taking the time to review our paper! Please find our responses below.
>
>
> - *The two-step formulation proposed for pipeline and data/tensor-model parallelism is novel and will greatly reduce the complexity of the overall search space.*
>
> We agree that this is a major contribution of our work and perhaps we could have emphasized this more in the paper. In fact, we hope that this work will spur the development of new algorithms for the compelling problem of automatically finding tensor-parallelism schemes for single layers, a problem which should be dramatically more tractable than trying to do the same for the entire DNN operator graph (combined also with pipelining and data parallelism).
>
>
> - *The algorithm assumes the knowledge of predefined “layers” and for every layer, there is a list of predefined model-parallel methods. For a more complicated neural network, this is not an easy task. This limits the application of the method on more complicated neural networks.*
>
> Indeed, the problem of finding good tensor-parallelism schemes for single layer types is beyond the scope of this paper, and it will require a different solution. However, at least the knowledge of "predefined layers" is not an issue - systems such as PyTorch allow one to export a layer (or "module") based computation graph.
>
>
> - *The paper only evaluates the BERT model. In this case, the DAG formulation actually degenerates to a chain, making the algorithm proposed in the paper overkill. The method should be evaluated on more neural networks with more complex architecture (that actually forms a DAG).*
>
> Thank you for this suggestion, which has prompted us to bolster our evaluation. Let us begin by remarking that our evaluations focused on huge Transformer-based language models such as BERT or GPT-3, as these form a large and very important class of models. It is this class that has seen exponential growth in recent years. In addition, their training is particularly expensive, which amplifies the benefits from optimal parallelization.
>
> Second, we would not say that the algorithm is overkill for DAGs that happen to be paths - it is still the right algorithm for the problem: we do not see any simpler solution for path-graphs, and there hasn't been any prior work that solves the problem (with data parallelism, tensor parallelism, etc.) for such graphs. There is also no overhead stemming from running a general-DAG algorithm on path graphs that could be avoided if the algorithm were tailored for path-graphs. In other words, given a path-graph, our general algorithm "degenerates" to the right algorithm for path-graphs.
>
> Third, while the focus of our evaluation was on Transformer-based language models, for more branching large DAGs that might arise in the future, we would in fact expect to see even larger gains - versus baselines that e.g. do equi-partitioning of the DNN - than we see in our BERT evaluations. To obtain a more complete picture, during the rebuttal period we have managed to source data for two more models, GNMT and Resnet50, profiled on nVidia GTX 1080Ti GPU, and evaluate Piper on them. (Note that GNMT is a DNN with high branching.) For both, we use multiple settings for the number of devices and the memory limit. We are happy to add these results to the final version. As expected, Piper remains efficient and outperforms the baselines. Here let us report some numerical highlights from the comparison with the baseline `equi`, which corresponds to the planner from the prior work PipeDream-2BW (equi-partitioning layers among the stages, trying all possible numbers of stages, data-parallel degrees, and enabling or disabling activation recomputation). For GNMT, the throughput of `equi` was on average 68% of that of Piper, and in one scenario `equi` could not find any memory-feasible solution while Piper was able to. For Resnet50, the throughput of `equi` was on average 46% of that of Piper, and we found more scenarios where `equi` could not find any feasible solution.
>
>
> - *Although Figure 4 partly proves the simulation is realistic, I’m still not convinced and I would like to see larger-scale experiments.*
>
> We were fortunate to get access to a 64-GPU nVidia A100 system for the submission, which allowed us to empirically confirm and demonstrate that the predicted and measured throughputs very closely match. We did so at the largest scale we could get our hands on (even if it was for a short duration). Given this close correspondence, we have concentrated on performing experiments using the predicted throughput. We believe that this is sound. The entire goal of showing such a correspondence was to bolster the confidence in the predicted throughputs we present in the rest of the paper. We would also like to note that it is not practical for us to e.g. obtain access to a system with 2048 GPUs. An advantage of our approach is that it does not require utilizing the large number of resources that would otherwise have been used for training -- thus being resource and energy footprint conscious for this work.
>
> A final thing to note is that the experimental setup from Figure 4 corresponds to the one for Figure 1a (for k=64). While in Figure 1a we do not plot the results for k=64, they look like an average of those for k=32 and k=128.
>
>
> - *Some key concepts will be easier to be understood with examples (e.g. Downsets). Line 26: Maybe also add \emph for “pipelined model parallelism”?*
>
> Good points. In fact we have figures to illustrate this, and we will include them in the final version (in the main body or in the appendix, depending on space constraints).
>
>
> - *Do you profile the Time-Per-Sample for every downset? How long does this take? Will this be a potential bottleneck when the neural network structure getting more complex?*
>
> No, in fact one only needs to profile the Time-Per-Sample for every node (which represents a layer), and this could even be reused between different nodes that correspond to the same layer type. This quantity is encoded as X.p in the TMPC X (see line 219 on page 5), which is a feature of a node.

---

> > ### Comment · Reviewer_BM6L · 2021-08-28
> > **Thanks**
> >
> > Thanks for the response! The response addresses my concerns and it would be great to include these into the final version.

---

### Decision · Program_Chairs · 2021-09-27

**Decision:**

Accept (Poster)

**Comment:**

After some discussion, there's general consensus toward accepting the work. As Reviewer cpuf notes: "It is the first work that uses dynamic programming to search in a non-trivial search space that includes data parallelism, tensor model parallelism, pipeline model parallelism, and activation rematerialization." There are a number of challenges in finding parallelizable configurations, and while the paper's specific ideas may have overlap with prior works or may be argued as incremental, it makes an important delta in solving this important problem area.

Flaw-wise, there are several experimental limitations that would be worth resolving as Reviewer BM6L and cpuf note. I highly recommend the authors address the rebuttal concerns as they further polish the paper.